



# Towards continuous monitoring of aerosol hygroscopicity by Raman lidar measurements at the EARLINET station of Payerne

Francisco Navas-Guzmán[1], Giovanni Martucci[1], Martine Collaud Coen[1], María José Granados-Muñoz[2], Maxime Hervo[1], Michael Sicard[2,3], and Alexander Haefele[1,4]

[1]Federal Office of Meteorology and Climatology MeteoSwiss, Payerne, Switzerland
[2]Remote Sensing Laboratory/CommSensLab, Universitat Politècnica de Catalunya, 08034 Barcelona, Spain
[3]Ciències i Tecnologies de l'Espai - Centre de Recerca de l'Aeronàutica i de l'Espai / Institut d'Estudis Espacials de Catalunya (CTE-CRAE / IEEC), Universitat Politècnica de Catalunya, Barcelona, 08034, Spain
[4]Department of Physics and Astronomy, The University of Western Ontario, London, Canada

**Correspondence:** Francisco Navas Guzmán (francisco.navas@meteoswiss.ch)

**Abstract.** This study focuses on the analysis of aerosol hygroscopicity using remote sensing technique. Continuous observations of aerosol backscatter coefficient, temperature and water vapour mixing ratio are performed by means of a Raman lidar system at the aerological station of MeteoSwiss at Payerne (Switzerland) since 2008. These measurements allow us to monitor in a continuous way any change of aerosol properties as a function of the relative humidity (RH). These changes can be observed either in time at constant altitude or in altitude at a constant time. The accuracy and precision of RH measurements from the lidar have been evaluated using the radiosonde (RS) technique as reference. A total of 172 RSs were used in this intercomparison which revealed a small bias ($< 4\%$RH) and standard deviation ($< 10\%$RH) in the whole troposphere between both techniques indicating the good performance of the lidar for characterizing RH. A methodology to identify aerosol hygroscopic conditions has been established and the aerosol hygroscopicity has been characterized by means of the backscatter enhancement factor ($f_\beta$). Two case studies, corresponding to different types of aerosol are used to illustrate the potential of this methodology. The first case corresponds to a mixture of rural aerosol and smoke particles which showed a higher hygroscopicity ($f_\beta^{355} = 2.8$ and $f_\beta^{1064} = 1.8$ in the RH range 73% - 97% ) than the second case, in which mineral dust was present ($f_\beta^{355} = 1.2$ and $f_\beta^{1064} = 1.1$ in the RH range 68% - 84%). The higher sensitivity of the shortest wavelength to hygroscopic growth is qualitatively reproduced using Mie simulations. In addition, a good agreement was found between the hygroscopic analysis done in the vertical and in time for case I, where the latter also allowed to observe the hydration and dehydration of this type of aerosol. Finally, the impact of aerosol hygroscopicity on the Earth's radiative balance has been evaluated using the GAME (Global Atmospheric Model) radiative transfer model. The model showed a significant impact with an increase in absolute value of 2.4 W/m$^2$ at the surface with respect to the dry conditions for the hygroscopic layer of case I (with presence of smoke).



# 1 Introduction

Atmospheric aerosol particles scatter and absorb solar radiation and therefore have an impact on the Earth's radiative budget (direct effect). In addition, aerosol particles can act as cloud condensation nuclei (CCN) and modify cloud microphysical properties altering also in this way the global radiative budget (indirect effects) (Haywood and Boucher, 2000). The uncertainty in

assessing total anthropogenic greenhouse gas and aerosol impacts on climate must be substantially reduced from its current level to allow meaningful predictions of future climate. This uncertainty is currently dominated by the aerosol component (Bindoff et al., 2013). Evaluation of aerosol effects on climate must take into account high spatial and temporal variation of aerosol concentrations and properties as well as the aerosol interactions with clouds and its influence on precipitation. During the last years a huge effort has been made in order to characterize vertically-resolved profiles of optical and microphysical

properties for different kind of particles. Raman lidars (light detection and ranging) have proven to be an essential tool to obtain profiles of these properties without modifying the environmental conditions (Ansmann et al., 1992a; Navas-Guzmán et al., 2013; Granados-Muñoz et al., 2014). Networks as EARLINET (European Aerosol Research Lidar Network) have contributed to create a quantitative, comprehensive, and statistically significant database for the horizontal, vertical, and temporal distribution of aerosols on a continental scale (Bösenberg et al., 2003; Pappalardo et al., 2014; Sicard et al., 2015).

However, despite of the big effort of the scientific community to characterize aerosol effects, there are still some processes that are not well understood yet. For example, an important factor that can modify the role of aerosols in the global energy budget is RH. Under high RH conditions, aerosol particles may uptake water changing their size and chemical composition (hygroscopic growth). This hygroscopic growth affects the direct scattering of radiation and especially the indirect effects, as the ability of aerosol to act as CCN is directly related to their affinity for water vapour (Hänel, 1976; Feingold and Morley,

2003; Zieger et al., 2013). Thus, understanding aerosol hygroscopic growth is of high importance to quantify the influence of atmospheric aerosol in climate models or for comparisons of remote sensing with in-situ measurements which are often performed under dry conditions (Zieger et al., 2010).

    Several studies have addressed the characterization of aerosol hygroscopicity using in-situ measurements over the last years. Techniques as the Humidified Tandem Differential Mobility Analyzer (HT-DMA) (e.g.Swietlicki et al. (2008)) or Humidified

tandem nephelometers have been extensively used to quantify the change in particle diameter or aerosol optical properties due to water uptake (Pilat and Charlson, 1966; Fierz-Schmidhauser et al., 2010; Zieger et al., 2013). However, despite of the great contribution that these techniques can provide to the understanding of the aerosol hygroscopic processes they also present some shortcomings. For example, most in situ techniques are limited by the fact that they modify the ambient conditions and are also subject to particle losses in the sampling lines, thereby altering the real atmospheric aerosol properties (Bedoya-Velásquez

et al., 2018). In addition, they present larger errors in the characterization of the aerosol hygroscopicity for high RH (conditions close to saturation), specially when the aerosol is more hygroscopic (Titos et al., 2016).

    In this sense, remote sensing could overcome theses difficulties since they can provide vertically resolved measurements without modifying the aerosol sample. In addition, this technique is able to measure under RH close to saturation, that is where particles are more affected by hygroscopic growth (Feingold and Morley, 2003). However, the number of aerosol hygroscopic



studies using remote sensing is modest and most of them were limited to specific field campaigns. The main limitation to address these studies comes from the difficulty to obtain RH profiles with high vertical and temporal resolution. Most of the studies carried out so far have used sensors to measure RH on board of RS or aircrafts or in meteorological towers in combination with the aerosol measurements from lidars to investigate the aerosol hygroscopicty (Wulfmeyer and Feingold, 2000;

Veselovskii et al., 2009; Granados-Muñoz et al., 2015; Haeffelin et al., 2016; Lv et al., 2017; Zhao et al., 2017; Fernández et al., 2018). Recently, the combination of temperature profiles from microwave radiometers (MWR) and water vapour mixing ratio ($r$) profiles from Raman lidar have been used to retrieve RH profiles (Navas-Guzmán et al., 2014) and applied to aerosol hygroscopic studies (Bedoya-Velásquez et al., 2018). However, although this synergy of instrumentation can be a good solution for many stations the uncertainties coming from MWR temperature profiles can be problematic due to its lower spatial resolu-

tion (Navas-Guzmán et al., 2016). In addition, most of the $r$ measurements from Raman lidar system are limited to night-time observations, due to the low signal-to-noise ratio of the Raman channels during daytime.

In the present study we show the capability of the RAman Lidar for Meteorological Observations (Ralmo) operated at the aerological station of MeteoSwiss at Payerne (Switzerland) to monitor aerosol hygroscopicity based on its continuous aerosol and relative humidity measurements. The methodology needed for this characterization is introduced and applied to two case

studies in which different aerosol types were present.

## 2   Experimental site and instrumentation

Data from collocated remote sensing and in-situ sensors were acquired at the aerological station of MeteoSwiss at Payerne (Switzerland, 46.82° N, 6.95° E, 491 m above sea level (asl)). The station is located in a rural area on the Swiss Plateau, between the Jura mountains (25 km to north-west) and the Alpine foothills (20 km to the south-east). The site is not significantly

affected by industrial pollution and the region is characterized by a mid-latitude continental climate.

Aerosol vertical information is mainly obtained from lidar systems installed at the station. The Raman Lidar for Meteorological Observations, Ralmo, is the main tool of this study. Ralmo was developed by the Swiss Federal Institute of Technology (EPFL) in collaboration with MeteoSwiss and is operated at MeteoSwiss Payerne since August 2008. The instrument is dedicated to operational meteorology, model validation, climatological studies as well as ground truthing of satellite data. The lidar

was specially designed to satisfy the requirements for operational monitoring of the atmosphere and to create a durable and homogeneous dataset to be used for climatology studies (Dinoev et al., 2013). The lidar system uses a Nd:YAG laser source which emits pulses of 8 ns duration at a wavelength of 355 nm and with a repetition rate of 30 Hz. The mean energy per pulse at 355 nm is around 400 mJ. Before being emitted into the atmosphere the beam is expanded to a diameter of 14 cm which reduces the beam divergence (to 0.1 mrad) and ensures eye-safety. The receiving system consists of four telescope with

30-centimeter parabolic mirrors which are arranged symmetrically around the vertically mounted beam expander to receive the backscattered photons. The total aperture of this telescope system is 60 cm diameter and it has a field of view of 0.2 mrad. This narrow field of view combined with the narrow-band receiver and high pulse energy allows daytime operation. Optical fibers connect the telescope mirrors with two grating polychromators which allow to isolate the rotational-vibrational Raman



signals of nitrogen and water vapor (wavelengths of 386.7 and 407.5 nm, respectively) and pure rotational Raman lidar signals (around 355 nm) for temperature, aerosol backscatter and extinction measurements. The optical signals are finally detected by photomultipliers and acquired by a transient recorder (Brocard et al., 2013). A detailed description of the different parts of this lidar system can be found in Dinoev et al. (2013). Ralmo was incorporated to EARLINET in 2008.

In order to gain more spectral information in the profiles of aerosol properties, data from a co-located ceilometer has been used in this study. The ceilometer CHM15k from Lufft company operated at Payerne is a lidar cloud height sensor based on a single wavelength backscattering technique. Its Nd:YAG narrow-beam microchip laser operates at 1064 nm. Cloud layers can be detected in a range of up to 15 km. In addition to the cloud base information, we derive the aerosol backscatter coefficient at 1064 nm from the elastic signal using the Klett inversion technique (Klett, 1981).

RS measurements were also used in this study to assess the temperature, water vapour and relative humidity profiles retrieved from Ralmo. The SRS-C50 sondes have been flown operationally at Payerne and launched twice a day at 11 UTC and 23 UTC. The sondes were launched and transported through the troposphere and stratosphere by a balloon inflated with Hydrogen and reaching on average 35 km. The vertical resolution of the measured profiles of temperature, humidity and temperature is of about 6 m. The sensors of these RSs include copper–constantan thermocouples for temperature, a full range water hypsometer
for pressure and a sensor with hygristor for relative humidity. The accuracy of these three parameters in the troposphere is 0.1 K for temperature, 2 hPa (accuracy decreases with height) for pressure and 5 to 10% for RH.

    Sun-photometer measurements have also been included in this study to complement the aerosol information. MeteoSwiss operates four Precision Filter Radiometers (PFR) that were installed in the framework of the international GAW-PFR (Global Atmosphere Watch - Precision Filter Radiometer) program of the World Meteorological Organization (WMO). They measure
the direct solar irradiance at the four wavelengths (368, 412, 500 and 862 nm) used in the GAW PFR network as well as 9 additional wavelengths (305, 311, 318, 332, 450, 610, 675, 718, 778, 817, 946 and 1024 nm). Integrated water vapor (IWV) is obtained from the measurements at 718, 817 and 946 nm. The Aerosol Optical Depths (AODs) have been calculated based on the solar irradiance at the different wavelengths using the GAW-PFR algorithms (McArthur et al., 2003). In addition, the AOD Angström exponent (AE) is used in this study as a qualitative indicator of the relative dominance of fine and coarse mode
aerosols. Values of AE larger than 1.5 are indicative of a higher fine mode fraction (Nyeki et al., 2012).

    In-situ observations at ground-level were also used to complement the column aerosol information obtained from remote sensing. These in-situ measurements at Payerne are carried out by EMPA in the framework of the Nabel (National Air Pollution Monitoring Network) monitoring program. Aerosol absorption coefficients at 7 wavelenghts (from 370 to 950 nm) were obtained from aethalometer measurements (AE31 model). Light absorption measurements at ultraviolet, visible and infrared
wavelengths can be used to quantitatively assess the aerosol source contribution (Collaud Coen et al., 2004; Sandradewi et al., 2008; Segura et al., 2014). In addition, concentrations of PM10 and PM2.5 (ambient air levels of atmospheric particulate matter finer than 10 and 2.5 microns) were obtained from an aerosol spectrometer (Fidas 200). These measurements were used to characterize the size of the particles that arrived to our station.



## 3 Methodology

### 3.1 Retrievals of relative humidity and aerosol property profiles

As mentioned in the introduction, the main practical limitation to study the effect of hygroscopicity on optical and microphysical aerosol properties is the lack of simultaneously and continuously available aerosol and RH measurements with a good

vertical and temporal resolution. Ralmo can overcome this limitation thanks to its capability to perform continuous measurements of temperature, water vapour and aerosol profiles during day and night. In this section the methodology to retrieve these atmospheric parameters from the lidar signals is described.

Temperature measurements are made using the pure-rotational Raman (PRR) technique (Vaughan et al., 1993). An atmospheric temperature profile can be derived from the analysis of the intensities of lines with low and high quantum numbers that

have opposite temperature derivatives. A detailed description of the temperature inversion technique applied to our system can be found in Martucci et al. (in preparation).

The mixing ratio, $r$, is defined as the ratio of the mass of water vapour to the mass of dry air in a sample of the atmosphere (Goldsmith et al., 1998). Profiles of $r$ can be obtained from Raman lidar measurements as the ratio of rotational-vibrational Raman scattering intensities from water vapour and nitrogen molecules (Whiteman, 2003a, b; Brocard et al., 2013; Navas-

Guzmán et al., 2014). It can be expressed as follows:

$$r(z) = C \frac{S_{H_2O}}{S_{N_2}} \exp \left\{ \int\limits_0^z \left[ \alpha \left( r, \lambda_{H_2O} \right) - \alpha \left( r, \lambda_{N_2} \right) \right] dr \right\} \tag{1}$$

where $C$ is the lidar calibration coefficient that takes into account the fractional volume of nitrogen in the atmosphere (0.78), the instrumental transmission and detection efficiencies at the wavelengths of the Raman returns and the range-independent Raman backscatter cross section for nitrogen and water vapour. The calibration coefficient must be determined for each specific lidar.

In the case of Ralmo $C$ is obtained combining absolute calibration with co-located radiosounding, with relative calibration using the solar background (Martucci et al., 2018). $S_{H_2O}$ and $S_{N_2}$ are the Raman lidar signals for water vapour and nitrogen, respectively, after being corrected for saturation and background. The exponential term is the difference in the atmospheric transmission between the surface and the altitude z for nitrogen (386.7 nm) and water vapour (407.5 nm). The molecular extinction can be calculated considering US Standard Atmosphere. For normal conditions at Payerne the effect of differential

extinction due to aerosols between these two wavelengths is small and can be considered negligible (Dinoev et al., 2013). However, when this differential extinction is not negligible, Muñoz-Porcar et al. (2018) showed that it can accurately be calculated using a radiative transfer model with a relatively simple parametrization.

RH profiles are obtained combining temperature and $r$. RH is defined as the ratio of the actual amount of water vapour in the air compared to the equilibrium amount (saturation) at that temperature (Yau and Rogers, 1996), and it can be calculated as:

$$RH(z) = \frac{e(z)}{e_w(z)} \times 100, \tag{2}$$



where $e(z)$ is the water vapour pressure while $e_w(z)$ is referred to the saturation pressure. The water vapour pressure is related to $r$ as follows:

$$e(z) = \frac{p(z)r(z)}{0.622 + r(z)}, \tag{3}$$

where $p(z)$ is the air pressure profile which can be estimated from RS or assuming a US standard atmosphere. In our case, and thanks to the availability of operational RS measurements in our station, the closest RS in time is used to calculate the pressure profile. For saturation pressure we use the following expression:

$$e_w(z) = 6.107 \exp\left[\frac{M_A\left[T(z) - 273\right]}{M_B + \left[T(z) - 273\right]}\right], \tag{4}$$

where the constants $M_A$ and $M_B$ are 17.84 (17.08) and 245.4 (234.2), respectively, for $T$ below (above) 273 K (List, 1951).

Aerosol vertical information is also retrieved from Ralmo measurements. The backscattered radiation from aerosols is due to Mie scattering and it has the same wavelength as the laser emission (no wavelength shift). The Rayleigh scattering from molecules presents also the same wavelength as the incident radiation. The sum of both signals (molecules plus particle) is called elastic signal, $S_{El}(z)$. Stokes and anti-Stokes portions of the PRR spectrum with opposite temperature dependence are summed, giving in good approximation a temperature independent signal $S_{PRR}(z)$. The particle backscatter coefficient, $\beta_{aer}$, can be derived from the ratio of the total (particle plus molecular) and pure molecular backscatter signals (Ansmann et al., 1992b). In the case of Ralmo, $\beta_{aer}$ is deduced taking the ratio of the elastic backscatter $S_{El}(z)$ (Mie and Rayleigh scatter) to the PRR signal $S_{PRR}(z)$ as follows:

$$\beta_{aer}(z) = \beta_m(z)\left[k\frac{S_{El}(z)}{S_{PRR}(z)} - 1\right], \tag{5}$$

where $k$ is the calibration constant and $\beta_m$ is the molecular backscatter coefficient. $\beta_m$ can be calculated using a modeled or measured atmospheric density while the $k$ constant can be estimated assuming a molecular behaviour of the atmosphere in the far range (upper troposphere).

### 3.2  Selection of aerosol hygroscopic cases

Continuous measurements of aerosol and RH profiles from Ralmo lidar allow us to monitor any change in aerosol properties that could occur as result of the water uptake by the particle under high RH (aerosol hygroscopic growth). However, to ensure that the changes in the aerosol properties are due to hygroscopic growth and not to changes in the load or composition of the aerosol certain requirements must be fulfilled.

As a first condition an increase in the aerosol particle backscatter should occur simultaneously with an increase in RH. This condition could be observed in the vertical for a certain aerosol layer (vertical hygroscopic growth) but it could also be found as a function of time at a defined altitude. It is worth to remark that Ralmo is one of the few remote sensing instruments in the world that is able to monitor those processes as a function of time and height. Cases fulfilling the previous condition are selected as potential cases of hygroscopic growth. After that, we need to ensure a high degree of homogeneity in the investigated aerosol layer. For that, a second requirement to be fulfilled is that the origin of the air masses, in the region where the presence





of hygroscopicity is assessed, is independent of the altitude (in case of vertical hygroscopic growth). To guarantee that we have used backward trajectory analysis from HYSPLIT model (Hybrid Single-Particle Lagrangian Integrated Trajectory) (Draxler and Rolph, 2003). As final condition to ensure a good mixing within the layer we use the profiles mixing ratio and potential temperature obtained from Ralmo measurements. Slow-varying or constant values of these two parameters are indicative of

well-mixed conditions across the aerosol layer.

Similar criteria to assess the homogeneity of an aerosol layer have been required in previous studies using remote sensing (Veselovskii et al., 2009; Granados-Muñoz et al., 2015; Bedoya-Velásquez et al., 2018). However, this is the first time that the profiles of mixing ratio and potential temperature are obtained from the same instrument as the aerosol measurements. It presents a clear advantage since all the parameters are measured for the same atmospheric column as opposed to studies using

RSs or MWR.

An other novelty of this study is the capability to monitor aerosol hygroscopic growth in time at different altitudes in the troposphere. For this temporal analysis we have used the mixing ratio measurements to ensure that changes of aerosol properties in time are only due to hygroscopic processes. We assume that the mixing ratio is a good atmospheric tracer and we consider that small variations in time indicate no significant change in the origin of the air mass. In addition, wind measurements from

the wind profiler are used to corroborate that the direction of the air mass was the same during the analyzed period.

Once all the previous requirements have been fulfilled the aerosol hygroscopicity is characterized by means of the enhancement factor. This parameter is defined as

$$f_\zeta(RH) = \frac{\zeta(RH)}{\zeta(RH_{ref})} \tag{6}$$

where $\zeta(RH)$ represents a determined aerosol property at a certain RH. $RH_{ref}$ is the so-called reference RH and it is chosen

as the lowest value of RH in the analyzed layer or time interval. In this study, the enhancement factor has been calculated for the aerosol backscatter coefficient.

In order to be able to compare our results with other studies in which $RH_{ref}$ or the RH range could be different the humidograms (RH versus the enhancement factors) are parameterized using fitting equations (e.g. Titos et al. (2016)). In this study we use the one-parameter equation introduced by Hänel (1976), which has been used in other hygroscopic studies using remote

sensing (Granados-Muñoz et al., 2015; Bedoya-Velásquez et al., 2018). The general form of of the Hänel equation is expressed as

$$f_\zeta(RH) = \left( \frac{1 - RH/100}{1 - RH_{ref}/100} \right)^{-\gamma} \tag{7}$$

where $\gamma$ is an indicator of the aerosol hygroscopicity. Larger values of $\gamma$ indicate higher hygroscopicity.

## 4   Validation of lidar measurements versus operational RSs

Since data quality is critical a validation with respect to the RS technique has been carried out, which we consider the reference. As it was indicated in Section 2, operational RS are launched twice per day at the aerological station of Payerne. The availability



of simultaneous lidar and RSs measurements at our station allows us to minimize the differences due to spatio-temporal inhomogeneities.

Figure 1 shows the mixing ratio, temperature and RH profiles obtained from Ralmo lidar and RS at nighttime (23:00 UTC) on 20th September 2017. A very good agreement can be observed from this figure for the three atmospheric parameters. We

can see that Ralmo provided very accurate results even for altitude ranges where strong gradients were observed, as for the wet layer located above 2 km above ground level (agl) or for the temperature inversion observed at 1.7 km (agl). Mean and standard deviation for the whole profile (from ground to 8 km (agl)) of the relative differences in $r$ were $-1 \pm 13\%$, while mean and standard deviation of temperature and RH differences were -0.1$\pm$0.7 K and -1$\pm$6 %RH, respectively. This example shows the potential of Ralmo to provide accurate measurements with a good spatial resolution.

In order to evaluate the accuracy and the precision of Ralmo retrievals a statistical analysis of lidar and RS differences has been carried out. A total of 172 RS were used in this intercomparison during day- and night-time for the period from July to December of 2017. The temperature validation of Ralmo lidar has been treated in more depth in a separate paper Martucci et al. (in preparation). Here we would only like to discuss the precision and accuracy of Ralmo retrieving this property based on the mean and standard temperature deviations for this 6-month statistics. For night-time measurements (23:00 UTC) a total

of 100 RS were used to compare with lidar and the mean temperature deviation profile evidenced that there was almost no bias between both techniques in the whole profile (mean temperature deviations of $0.05 \pm 0.06$ K in the first 5 km and $0.15 \pm 0.15$ K above that altitude). The standard deviation profile also confirmed the excellent performance of Ralmo retrieving temperature with standard deviations below 1 K in the full troposphere (mean values of $0.6 \pm 0.1$ K below 5 km (agl) and $1.00 \pm 0.16$ K above). For daytime measurements (11:00 UTC) 72 pairs of profiles were used. The mean temperature deviations between

lidar and RS for daytime measurements were of $-0.5 \pm 0.2$ K in the lower troposphere (0-5 km agl) and -0.1$\pm$0.6 in the upper troposphere (5-10 km agl). The temperature standard deviations were $0.8 \pm 0.2$ and $2.4 \pm 0.8$ for the same two altitude ranges. These results prove also the good performance of Ralmo during daytime, although showing larger discrepancies than during nighttime, especially in the upper troposphere (from 5 to 10 km). The reliability and the high quality of the temperature profiles obtained by Ralmo is a key aspect for addressing aerosol hygroscopic studies, since this is the most difficult to obtain of the

atmospheric parameters using remote sensing techniques.

Figure 2 presents the same statistics than in the previous discussion but for RH measurements. As it was explained in Section 3, RH profiles from lidar were obtained from the combination of mixing ratio and temperature measurements. Figure 2a shows all RH deviation profiles between lidar and RS while Figure 2b presents the mean RH deviation profile. This plot reveals a small bias between both instruments that ranges from positive values (+3% at 1.4 km asl) to negatives (-9% at 5.6 km asl). Above

9 km (asl) the bias becomes again positive reaching a maximum value of +6 %RH. The mean bias value in the lower region where the lidar measures larger values of RH than RS was +2.0$\pm$0.9 %RH (from ground to 2.1 km asl) while the mean bias for range 2.1-9 km asl was -4$\pm$2 %RH. We can affirm that the shape observed in the RH bias between both instruments is mainly coming from the mixing ratio measurements since the temperature statistics showed a negligible bias in the whole column (with almost zero bias, not shown). It is due to larger inhomogeneities in mixing ratio in time and in space which produces

larger discrepancies between lidar and RS observations. Regarding the standard deviations observed for this parameter (Fig.





2c) we can observe that it increases with the altitude in the lower troposphere (from 4.4%RH at ground to 8%RH at 2 km asl). Above this altitude the standard deviation values oscillate around a quite constant value in altitude. The mean RH standard deviation in the lower troposphere (from ground to 2 km) was 6±1%RH while it was 8±2%RH above this altitude. The RH comparison for daytime measurements (not shown here) also presented a small bias between lidar and RS, with a mean values

of -0.4±2.4 %RH in the range from ground to 5 km asl. The mean standard deviations obtained for the same altitude range was slightly larger than during nighttime with a mean value of 9±3 %RH. Above 5 km (asl) the errors were much larger due to the solar background radiation and RH profiles were not calculated above this altitude to avoid these large uncertainties. In any case it is important to point out the good accomplishment of Ralmo for retrieving RH information, as it can be concluded from this intercomparison in which Ralmo showed very small biases and standard deviations (below 9%RH) which is indicative of

the accuracy and the precision, respectively, of our measurements.

The good performance of Ralmo lidar retrieving RH measurements proved in this intercomparison is a key aspect to be able to address the aerosol hygroscopic studies as will be shown in the next sections.

## 5   Study of aerosol hygroscopicity

Two case studies are presented in which the hygroscopicity of different types of aerosol is characterized. The two case studies

took place during summer 2017 and there was presence of smoke particles in one of them and mineral dust in the other one.

### 5.1   Case I: hygroscopicity in smoke particles

Case I corresponds to 7 September 2017. The temporal evolution of $r$, RH and $\beta_{355}$ in the lower troposphere (0-5 km asl) is shown in Fig. 3 for this day. According to the aerosol measurements (lowest pannel), low clouds were present at around 1.6 km (agl) during the first part of the day (until 12 UTC). After that, two clear aerosol layers can be identified, the planetary

boundary layer (PBL) and a strong lofted aerosol layer located between 2 and 4 km (agl) which appeared after noon. An usual convective boundary layer pattern was observed along the day with maximum height occurring at around 14:00 UTC. It is interesting to remark how within the PBL the intensity of the aerosol backscatter signal was stronger at the top of this layer even when a strong mixing was expected at the central hours of the day (between 13:00 and 17:00 UTC). However, a larger homogeneity is observed for the same layer in the $r$ measurements (Fig. 3, upper pannel), indicating that the convective

processes were strong enough to produce a well mixed PBL for that time interval. The third element, that can help to understand the observed variations in the backscatter signals is the RH measurements (central panel). From that plot, we can observe how the intensification observed in the aerosol backscatter is well correlated with the values observed for RH, with the highest values of both properties at the top of the PBL. This kind of behaviour could be due to aerosol hygroscopic processes and a detailed analysis is presented in the following paragraphs.

According to the NAAPS model, 7 September 2017 at 18:00 UTC is characterized by the presence of smoke in our station (Fig. 4a). This model predicted smoke surface concentrations between 4 and 8 $\mu g/m^3$ over Payerne. The Hysplit backtrajectories analysis indicated that the air masses above our station in the lower troposphere (from ground to 3 km asl) had their





origin in North America (Fig. 4b). The VIIRS (Visible Infrared Imaging Radiometer Suite) fire and thermal anomalies product available from the joint NASA/NOAA Suomi-National Polar orbiting Partnership (S-NPP) satellite (Fig. 4c), showed that for the studied period (from 25 August to 3 September of 2017) several intensive hot spots were found along the calculated air mass trajectories, especially in some areas of Northwest of the United States and in the central part of Canada. In addition,

Carbon Monoxide (CO) observations from the Atmospheric Infrared Sounder (AIRS) on board of Aqua satellite monitored a plume (not shown here) with high concentration of CO that moved from North America to Europe during that period, reaching the Eastern part of Europe on 6 September 2017. Over Payerne, the total column CO concentrations observed with AIRS were in the range of 100 and 130 parts per billion by volume (ppbv) during the days 7 and 8 of September, which are considerably higher than the mean concentration observed in the previous month (70-80 ppbv). CO is considered as good tracer of smoke

particles since it is generated in the incomplete combustion of biomass burning.

In-situ measurements carried out at Payerne station showed some changes in the aerosol properties that could be characteristics of smoke particles (Fig. 5). An increase in the PM2.5 mass concentration was observed in that days with values changing from 1 $\mu g/m^3$ (12 UTC on 6th September) to 9.9 $\mu g/m^3$ (20 UTC on 8th September), indicating an increase in the concentration of small particles at the surface. The PM2.5 concentrations reached during that evening were clearly above than the

mean summer value (5.2 $\mu g/m^3$). The PM2.5 increase took place at the same time of an increase of the absorption coefficient at different wavelengths observed by an aethalometer. The absorption Angstrom exponent (AAE) which represents the wavelength dependence of absorption and depends on the composition of absorbing aerosols showed relatively low values (between 1.1 and 1.3 for most of the measurements) which can be characteristics of biomass burning particles (Schmeisser et al., 2017). Therefore, model predictions and satellite and in-situ observations agreed and pointed out to a mixture of local aerosol and

smoke particles from biomass burning in the atmospheric column of Payerne.

Vertical information of aerosol, temperature and mixing ratio has been obtained using the Ralmo lidar and ceilometer measurements on 7 September 2017. Figure 6 shows the profiles of $\beta$ at 355 nm and 1064 nm, RH and the auxiliary information of potential temperature and mixing ratio for the time interval 15:00-15:30 UTC. From this figure, a marked increase of $\beta$ with altitude is observed for the altitude range between 1.7 and 2.3 km (asl). Simultaneously to this increase, we can observe

that there is also an increase in RH for the same layer with values moving from 73% (bottom of the layer) to 97% (top of the layer). The profiles of potential temperature and mixing ratio (Fig. 6c) are used as indicators of a good mixing as explained in section 3.2. These profiles show quite constant values for both properties within the layer (300.6 ± 0.5 K and 5.3 ± 0.1 g/kg, respectively) indicating that the layer is well mixed.

Following the methodology presented in Sect. 3.2 the enhancement factors of the aerosol backscatter coefficients ($f_\beta(RH)$)

at 355 and 1064 nm have been obtained for the investigated layer from the combination of backscatter and RH measurements. Fig. 7 shows the dependency of $f_\beta(RH)$ with RH what is called humidogram. The reference RH for this case was 73% which corresponds to the lowest value in the layer. From this figure, we can observe how $\beta_{355}$ increased 2.8 times ($f_\beta(97\%)$=2.8) in the range of humidity between 73% and 97% (blue points). Lower values of the enhancement factor are observed at 1064 nm (red points). For this infrared channel the $\beta$ increased 1.8 times respect its value at $RH_{ref}$ (73%), indicating a lower sensitivity

of this wavelength to the aerosol hygroscopic growth. Hänel hygroscopic parameters ($\gamma$) obtained using the fitting equation 7



were calculated (solid lines) in order to make our measurements comparable with other studies. In this case, $\gamma$ took a value of 0.48 at 355 nm and 0.29 at 1064 nm. This parameter is proportional to the hygroscopicity observed in the aerosol property. Independent vertical profiles obtained from the measurements of the previous 30-min time interval (from 14:30 to 15:00 UTC) were also analyzed in order to check the consistency of our results (figure not shown here). For that period, a simultaneous

increase of aerosol backscatter and RH was also observed in the altitude range 1.5-2.2 km. Although the RH range observed for this time interval (69-95%) was slightly different from the one showed in Fig. 6, the Hänel parameters showed quite consistent results for both wavelengths ($\gamma_{355} = 0.57$ and $\gamma_{1064} = 0.35$). Similar values of hygroscopic parameter at 355 nm ($\gamma_{355} = 0.40$) have been reported by Bedoya-Velásquez et al. (2018) associated with the presence of smoke particles in which combination of lidar and MWR measurements were used for the aerosol hygroscopic analysis. However, the spectral dependency found in

our case is not what has been reported in other studies. A stronger efficiency for longer wavelengths (between 355 and 532 nm) have been found in $\gamma$ in other studies for smoke particles and also for anthropogenic particles using remote sensing (Bedoya-Velásquez et al., 2018; Lv et al., 2017). Haarig et al. (2017) also reported this higher efficiency to the hygroscopic growth at 1064 nm but for marine particles. In-situ studies have also showed this higher efficiency for longer wavelengths although over a shorter range (450 - 700 nm) on marine particles (Kotchenruther et al., 1999; Zieger et al., 2013).

We performed Mie simulations in order to understand if the spectral dependency observed in our study could be realistic or not. Backscatter coefficients at the lidar wavelengths (355, 532 and 1064 nm) as a function of RH were computed using a Mie code (Bond et al., 2006; Mätzler, 2002). For these simulations some inputs such as the aerosol growth factor (1.6 which is typical for hygroscopic aerosol) and the refractive index (m = 1.5 + i0.01) were assumed. Figure 8 shows the $f_\beta$ calculated from the Mie calculation as a function of RH for different wavelengths and particle sizes. Monomodal distributions were used

assuming different diameters for the dry particles and a geometric standard deviation of 1.5. From this figure we can observe how the backscatter is very sensitive to the wavelength and particle size as expected and whose relationship is characterized in scattering theory by the size parameter ($x = \pi D/\lambda$). Mie scattering regime is expected for x≈1. A stronger increase in the backscattered radiation at the shortest wavelengths is expected for small particles ($D_{dry}$=200 nm) when RH increases according to these simulations (Fig. 8,a). These results indicate that even a decrease in the backscatter at 1064 nm could be expected when

the size particle increase due to hygroscopic growth. Figure 8,b shows the humidogram for a dry particle diameter of 400 nm. A different spectral dependency is observed for this case with slightly higher $f_\beta$ at 532 nm than at 355 nm and lower at 1064 nm. These results agree with our observations and also explain the different spectral dependency observed in other studies for smoke and anthropogenic particles that could have similar sizes (small particles) to what is simulated here. Pannels c and d in Figure 8 show that for the biggest dry particle diameters (600 and 800 nm) the spectral dependency of $f_\beta$ is totally inverted

with respect to small particles with larger $f_\beta$ values at the longest wavelengths. This spectral dependency agrees with the observations in Haarig et al. (2017) for marine particles which are considered much bigger particles than smoke particles. However, we must also point out that for many cases larger particles in the atmosphere also means that we are further away from the ideality of a sphere considered in the theory of Mie, so this type of comparison should be considered with caution.

Continuous aerosol and RH measurements from Ralmo lidar also allows to monitor aerosol hygroscopic processes occurring

in time. Figure 9 shows the evolution of mixing ratio, wind direction, $\beta_{355}$ and RH at 1.3 km (asl) on 7 September 2017. As it





was indicated in Section 3.2, water vapour is considered a good tracer in the atmosphere and constant values of mixing ratio mean same origin of the air parcels. For this case, we can observe that $r$ (Fig. 9, top ) is quite constant (6.7±0.3 g/kg) during the evening (from 16:00 to 23:30 UTC), fulfilling the previous criterion. In addition, a simultaneous increase of $\beta_{355}$ and RH was observed for the indicated period and altitude (Fig. 9, bottom). The RH changed from 63% in late afternoon to reach values
close to 90% at midnight.

In order to quantify the aerosol hygroscopic effect that took place in time, $f_\beta$ was calculated again for these measurements (Fig. 10a, blue filled circles). The initial value of $\beta$ (at $RH_{ref} = 63\%$) increased by a factor of 2 when RH reached the maximum values of the evening. The Hänel parameterization was used for this dataset providing a hygroscopic parameter of 0.54 which is in very good agreement with the values observed in the atmospheric column in the afternoon of this day. In
addition to this hydration process (water uptake) we could also observe the dehydration (evaporation) suffered by this aerosol during the afternoon of this day (from 11:00 to 16:00 UTC, Fig. 9). In this period, $r$ and the wind direction measurements showed very stable values (6.0±0.5 g/kg and 302°± 27°, respectively), evidence that the air mass did not change either. A decrease of $\beta_{355}$ took place at the same time that RH decreased from 93% to 59%. The humidogram obtained for this dehydration process (Fig. 10a, blue open circles) also showed $f_\beta$ values very close to the ones calculated in the later period
with a hygroscopic parameter of 0.40. The same behaviour was also observed for $\beta_{1064}$ from the ceilometer measurements along this day, with hygroscopic parameter values of 0.41 and 0.34 for hydration and dehydration processes, respectively, showing again lower values than at the ultraviolet channel. We would like to remark the good agreement found in the aerosol hygroscopicity of this case using a vertical and a temporal analysis.

## 5.2 Case II: presence of mineral dust particles

The second case took place on 8 July 2017. Figure 11 (a) shows the evolution of mixing ratio, RH and aerosol backscatter at 355 nm from Ralmo during the morning of this day. From the aerosol measurements (lowest pannel) the presence of particles at high altitudes is evident, reaching almost 6 km (asl) in the late morning (around 9 UTC). The NMMB/BSC-Dust model (Pérez et al., 2011) predicted for this day presence of dust particles over the western part of Europe including Switzerland. A dust concentration profile calculated using this model in our Earlinet station (Fig. 11, b) showed higher concentration of
mineral dust in the lower troposphere with a profile very similar to what was observed with our Raman lidar (Fig. 11, c). Back-trajectories analysis from HYSPLIT model (not shown here) indicated that the observed air masses had their origin in North Africa. Remote and in-situ measurements carried out in our station also confirmed features typical of this kind of particles. AOD Angstrom exponent obtained from the PFR sun-photometer measurements presented low values (between 0.5 and 0.6) along the morning, indicating the presence of coarse particles in the atmospheric column. In-situ measurements also showed a
strong increase in the PM10 concentration at the surface along this day with values ranging from 14 $\mu g/m^3$ at 03:00 UTC to 37.4 $\mu g/m^3$ at 15:00 UTC (the annual mean PM10 concentration in 2017 was 12 $\mu g/m^3$).

Once the aerosol is well identified using models and measurements we analyze the vertical profiles obtained from the lidar (Fig. 12, left). We observe a simultaneous increase of $\beta$ from the lidar systems with the RH at the altitude range between 1.9 and 2.3 km (asl). For that range, mixing ratio and potential temperature showed quite constant values, evidence of a well





mixed layer. The dependence of backscatter enhancement factor at 355 and 1064 nm with the RH is shown in the resultant humidogram in Fig. 12 (right). Although there was an intensification of the backscatter in both wavelengths ($f_\beta^{355}(84\%) = 1.2$ and $f_\beta^{1064}(84\%) = 1.1$ with $RH_{ref} = 68\%$), it was much lower compared to case I. The hygroscopic parameter obtained from the Hänel parametrization confirmed this behaviour, with values of 0.20 and 0.12 at 355 and 1064 nm, respectively. These

values are similar to the ones observed by Lv et al. (2017) also under the presence of mineral dust ($\gamma_{355} = 0.12$ and $\gamma_{532} = 0.24$ ). However, we found again for this case that the spectral dependency is opposite compared to this other study, although in our case we work over a wider spectral range.

## 6    Evaluation of the effect of aerosol hygroscopicity on the Earth's radiative balance

Because of the changes in aerosol optical and microphysical properties due to water uptake, aerosol radiative properties are

also modified in case of hygroscopic growth. As stated before, the aerosol backscatter and extinction coefficients vary under high relative humidity conditions which in turn lead to an increase of the AOD. To compute the AOD the backscatter profiles at 355 nm were converted to extinction using a generic lidar ratio of 50 sr. In order to calculate $\Delta$AOD, the increment of AOD due to the hygroscopic growth, we obtain a so-called "dry" aerosol extinction profile by using the Hänel parameterization and assuming that RH in the analyzed layer is equal to $RH_{ref}$ for each case. The "dry" profiles obtained are included in Figure 13.

AOD for the dry and wet cases, as well as $\Delta$AOD, are summarized in Table 1.

For the two cases analyzed here, the increase in AOD at 355 nm related to the hygroscopic growth in the analyzed layers is $\Delta$AOD = 0.017 for Case I and $\Delta$AOD = 0.001 for Case II. In relative terms, this results in an increase of the total AOD of 4.7% (16.6% if we consider only the AOD in the PBL where the hygroscopic layer is located) due to hygroscopic growth for Case I, and 0.6% for Case II. As expected, $\Delta$AOD is much larger in Case I since smoke particles are more hygroscopic ($\gamma_{355}$

= 0.48) than mineral dust ($\gamma_{355}$ = 0.20). It is worth noting the significant effect of the hygroscopic growth on the PBL AOD which increases nearly 16%.

Simulations with a radiative transfer model can give us an estimate of the impact that this change in AOD has on the aerosol radiative effect (ARE). In this case, we use the GAME (Global Atmospheric ModEl, Dubuisson et al. (1996, 2005)) model. GAME is a modular radiative transfer model that allows calculating upward and downward radiative fluxes at different

vertical levels with high resolution by using the discrete ordinates method (Stamnes et al., 1988). Details about the model parameterization can be found in Sicard et al. (2014) and Granados-Muñoz et al. (2019). In our case, the variations in the ARE ($\Delta$ARE) in the shortwave spectral range are exclusively related to the variations in AOD and the AE due to hygroscopic growth, whereas the other aerosol properties are assumed to remain constant. The changes in AOD produce a net increase (in absolute values) of the aerosol radiative effect at the surface with respect to the "dry" profiles equal to 2.4 Wm$^{-2}$ (5.2%) and 0.1 Wm$^{-2}$

(0.4%) in Case I and Case II, respectively (for a solar zenith angle $\sim$30°). A solar zenith angle of 30° was chosen because it corresponds to the one of Case I. As expected, the effect is more noticeable in case of particles with stronger hygroscopic properties, such as the smoke in Case I. One also sees that the relative increase of AOD (4.7%) and ARE (5.2%) are similar. For the mineral dust event the effect is almost negligible ($\Delta$ARE=-0.1 Wm$^{-2}$). Variations of the ARE observed in previous studies





can reach up to 7 Wm$^{-2}$ (Stock et al., 2011), however a comparison with our data is not straightforward since these variations are highly dependent on the aerosol load and the aerosol type present in the atmosphere. In Case I, although the hygroscopic growth affects a quite thin layer (only 600 m width) and the $\gamma$ values are relatively low (0.48), the aerosol hygroscopic growth effect on the ARE is still quite noticeable. These results point out that in more favorable conditions, namely thicker layers

where the hygroscopic effect occur and particles with stronger hygroscopic properties, the aerosol hygroscopic effect on the optical and radiative properties could be quite considerable. In terms of radiative forcing efficiency, FE, the ratio of radiative forcing to AOD, the change from "dry" to "wet" induces a difference of -0.4 and +0.5 Wm$^{-2}$ in Case I and II, respectively, which in relative values represents less than 0.3% of the FE in the "dry" case. Such small changes suggest that the forcing efficiency seems to be insensitive to the hygroscopic growth for both smoke and dust particles. Anyhow, these results indicate

that including aerosol hygroscopic properties in climate model calculations is key for improving the accuracy of aerosol forcing estimates.

## 7    Conclusions

The present study demonstrates the capability of a Raman lidar to monitor aerosol hygroscopic processes. Continuous measurements of water vapour, temperature and aerosol profiles are performed by Ralmo lidar in almost a continuous way since

2008 at the aerological station of MeteoSwiss at Payerne (Switzerland). These measurements allow us to monitor any change in aerosol properties that could occur as result of the water uptake by the particle under high RH (aerosol hygroscopic growth). To ensure that the changes in aerosol are only due to hygroscopic growth several criteria have been established. As first condition an increase in the aerosol particle backscatter should occur simultaneously with an increase in RH. In addition, a high degree of homogeneity is required in the investigated layer. For that, backtrajectory analysis is used to verify that the origin of the air

mass is independent of the altitude and low-varying or constant values of mixing ratio and potential temperature are required as proxy for well-mixed conditions along the aerosol layer.

The accuracy and the precision of Ralmo temperature and RH profiles have been assessed using the reference RS technique. A total of 172 RSs were used in this intercomparison during day- and night-time for the period from July to December of 2017. The mean temperature deviations calculated from night-time (day-time) measurements revealed almost no bias between both

techniques in the whole troposphere, with mean temperature deviations of 0.05±0.06 K (-0.5±0.2 K) in the first 5 km and 0.15±0.15 K (-0.1±0.6 K) above that altitude. The standard deviations also confirmed the excellent performance of Ralmo retrieving temperature with values below 1 K in the full troposphere during nighttime and slightly larger values during daytime (0.8±0.2 K from ground to 5 km and 2.4±0.8 K from 5 to 10 km).

RH profiles from lidar were obtained from the combination of mixing ratio and temperature measurements. Small RH biases

were observed between both techniques during nighttime in the troposphere (from ground to 9 km asl) with values ranging between 3%RH at 1.4 km (asl) and -9%RH at 5.6 km (asl). The standard deviation of RH deviations also showed the good precision of the lidar measurements with values always lower than 9%RH. The RH performance of Ralmo during daytime were very similar to the ones obtained during nighttime in the lower troposphere (from ground to 5 km). However, above 5



km the errors were much larger due to the solar background radiation and RH profiles were not calculated to avoid these large uncertainties. The good performance of Ralmo lidar retrieving RH measurements found in this intercomparison is a key aspect to be able to address the aerosol hygroscopic studies.

The methodology presented here was applied to two case studies. In-situ and satellite measurements in addition to models indicate that Case I (7 September 2017) was characterized by a mixture of local aerosol and smoke particles from fires in North America. The enhancement factors of $\beta$ was found to be 2.8 at 355 nm and 1.8 at 1064 when RH increased from 73 % to 97% in the investigated layer. The Hänel hygroscopic parameter which is proportional to the aerosol hygroscopicity took values of 0.48 at 355 nm and 0.29 at 1064 nm in this case. Independent vertical profiles obtained for a previous time interval showed the consistency of our results. Other remote sensing studies have shown a larger sensitivity to hygroscopic growth for longer wavelengths for this type of particles in contrast to our results. However, those studies were carried out in a shorter wavelength range (355-532 nm). Haarig et al. (2017) also observed a higher hygroscopicity at 1064 nm than at 355 nm but for marine particles which are larger and much more hygroscopic than in our case. Mie simulations carried out in this study revealed that the spectral dependency of $f_\beta$ can change strongly depending on the particle size and the wavelength of the incident radiation supporting the spectral behaviour observed in our case.

The availability of continuous aerosol and RH measurements from Ralmo also allowed monitoring aerosol hygroscopic processes in time for this first case. The evolution of $\beta$ and RH at 1.3 km (asl) on 7 September 2017 showed two periods in which there was a simultaneous decrease of both parameters (dehydration process) followed by a simultaneous increase (hydration process). The Hännel hygroscopic parameters calculated from the humidogram ($f_\beta$ vs. RH) for both periods took values of 0.54 (0.41) and 0.40 (0.34) at 355 nm (1064 nm) for the hydration and dehydration processes, respectively, which are in very good agreement with the results obtained in the vertical analysis.

The aerosol hygroscopicity of a second case (8 July 2017) characterized by the presence of dust particles was also analyzed in this study. The vertical analysis of the lidar measurements revealed also hygroscopic growth for these particles but with a very different behaviour. A much lower hygroscopicity than for the previous case (with presence of smoke) was observed with $\beta$ increasing only 1.2 and 1.1 times at 355 and 1064 nm, respectively, when RH increased from 68% to 84%. The hygroscopic parameters obtained from the humidogram were 0.20 and 0.12 at 355 and 1064 nm, respectively, showing a good agreement with the values observed in other studies for mineral dust. It was also remarkable the lower spectral sensitivity to the aerosol hygroscopicity found for this type of particles.

Finally, the impact of aerosol hygroscopicity on the Earth's radiative balance has been evaluated for the two presented cases using a radiative transfer model (GAME). The aerosol hygroscopic growth in the investigated layers produced an increase in the AOD at 355 nm of 0.017 (4.7%) for the case with presence of smoke particles and 0.001 (0.6%) for the case with mineral dust. These changes in AOD produced a net increase (in absolute values) of the radiative effect at the surface of 2.4 $\mathrm{Wm}^{-2}$ (5.2%) and 0.1 $\mathrm{Wm}^{-2}$(0.3%) in Case I and Case II, respectively. The results were significant for the case with presence of smoke particles (more hygroscopic) despite the aerosol load of the investigated layer was not very high. Therefore we can conclude that the effect of aerosol hygroscopicity on optical and radiative properties can be quite considerable and have to be considered in climate model calculations for improving accuracy of aerosol forcing estimates.





In future work we want to exploit the large data set (10 years) of simultaneous aerosol and humidity profiles from this Raman lidar to carry out a statistical analysis of aerosol hygroscopic properties.

*Data availability.* Data used in this paper are available upon request from corresponding author (francisco.navas@meteoswiss.ch).

*Competing interests.* The authors declare that they have no conflict of interest.

*Author contributions.* FNG analyzed the data and wrote the manuscript, GM worked on the temperature retrievals, MH and MCC performed the Mie simulations, AA designed the experiment and MJGR and MS performed the calculations with the radiative transfer model (GAME). All authors provided comments on the manuscript.

*Acknowledgements.* This work was supported by the Swiss National Science Foundation trough project PZ00P2 168114. We thank to EMPA and the Swiss Federal Office for the Environment (FOEN) for providing the data of the in-situ measurements carried out at Payerne withing the Nabel monitoring program. We also acknowledge the financial support by the European Union's Horizon 2020 research and innovation program through project ACTRIS-2 (grant agreement no. 654109). The radiative transfer simulations performed with GAME are supported by the European Union MSCA-RISE Action (grant agreement no. 778349); the Spanish Ministry of Economy and Competitiveness (ref. TEC2015-63832-P) and EFRD (European Fund for Regional Development); the Spanish Ministry of Science, Innovation and Universities (ref. CGL2017-90884-REDT); the Unity of Excellence Maria de Maeztu (ref. MDM-2016-0600) financed by the Spanish Agencia Estatal de Investigación. This work was also supported by the Juan de la Cierva-Formación program (grant FJCI-2015-23904). The authors also kindly acknowledge Philippe Dubuisson (Laboratoire d'Optique Atmosphérique, Université de Lille, France) for the use of the GAME model.





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





**Table 1.** AOD, ARE and FE and for both dry and wet conditions and their difference. All relative values are given with respect to the dry conditions.

| | $\gamma_{355}$ | Layer width (m) | $AOD_{355}$ | $AOD_{355}^{dry}$ | $\Delta AOD_{355}$ | ARE (Wm$^{-2}$) | $ARE_{dry}$ (Wm$^{-2}$) | $\Delta$ARE (Wm$^{-2}$) | FE (Wm$^{-2}$) | $FE_{dry}$ (Wm$^{-2}$) | $\Delta$FE (Wm$^{-2}$) |
|---|---|---|---|---|---|---|---|---|---|---|---|
| Case I (smoke) | 0.48 | 600 | 0.379 (0.126)* | 0.362 (0.109)* | 0.017 (4.7%) (15.6%)* | -48.8 | -46.4 | -2.4 (5.2%) | -128.6 | -128.2 | -0.4 (0.3%) |
| Case II (dust) | 0.20 | 400 | 0.157 | 0.156 | 0.001 (0.6%) | -27.3 | -27.2 | -0.1 (0.4%) | -173.9 | -174.4 | 0.5 (0.3%) |

*Indicates values within the PBL (between the surface and 2.5 km)



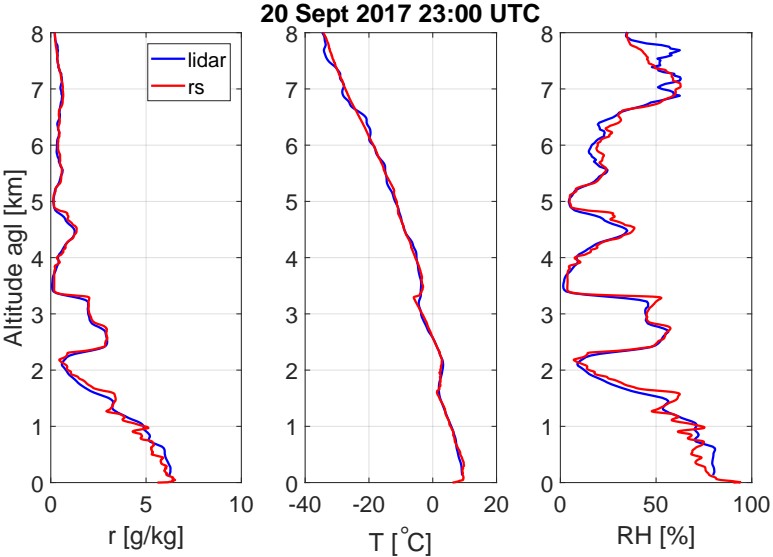

**Figure 1.** $r$, temperature and RH profiles from lidar (blue lines) and operational RS (red line) on 20th September 2017 at 23:00 UTC.

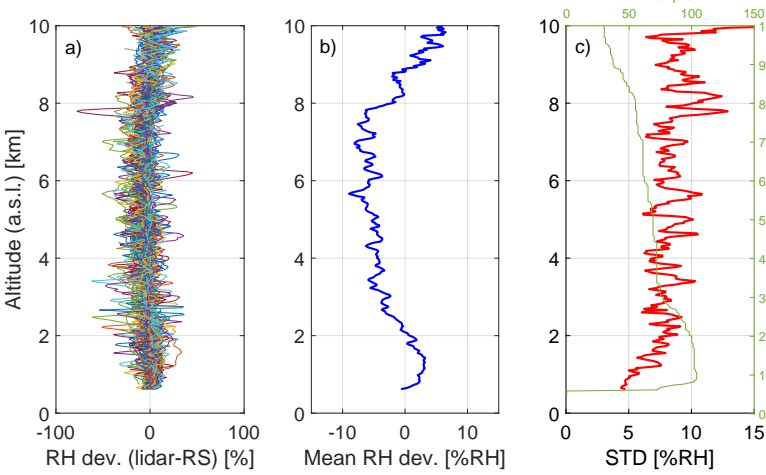

**Figure 2.** RH validation between Ralmo lidar and RS at 23:00 UTC. (a) Profiles of RH deviation between lidar and RS. (b) Mean RH deviation (lidar-RS). (c) Standard RH deviation profile (red line) and number of profiles used at each altitude for this statistics (green line).




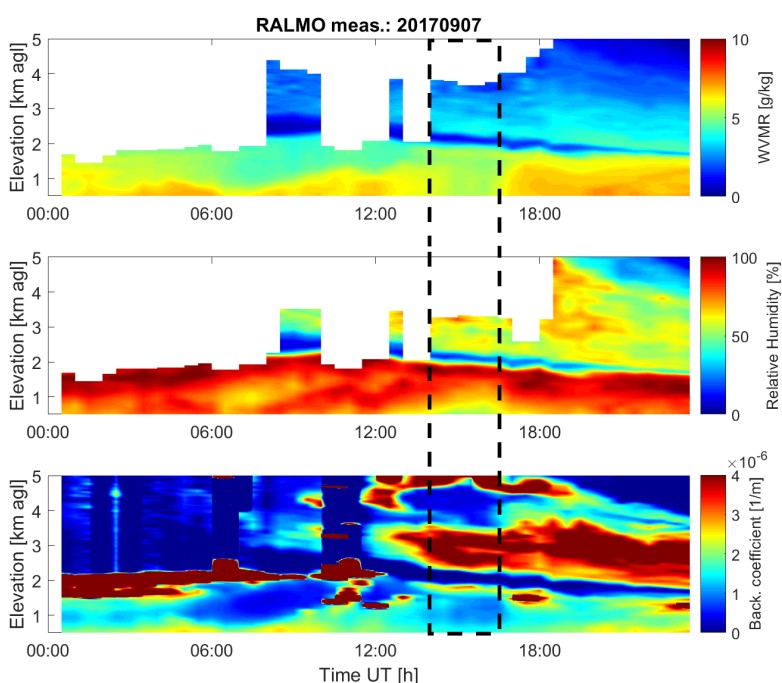

**Figure 3.** Temporal evolution of water vapour mixing ratio, RH and aerosol backscatter coefficient at 355 nm on 7th September 2017.



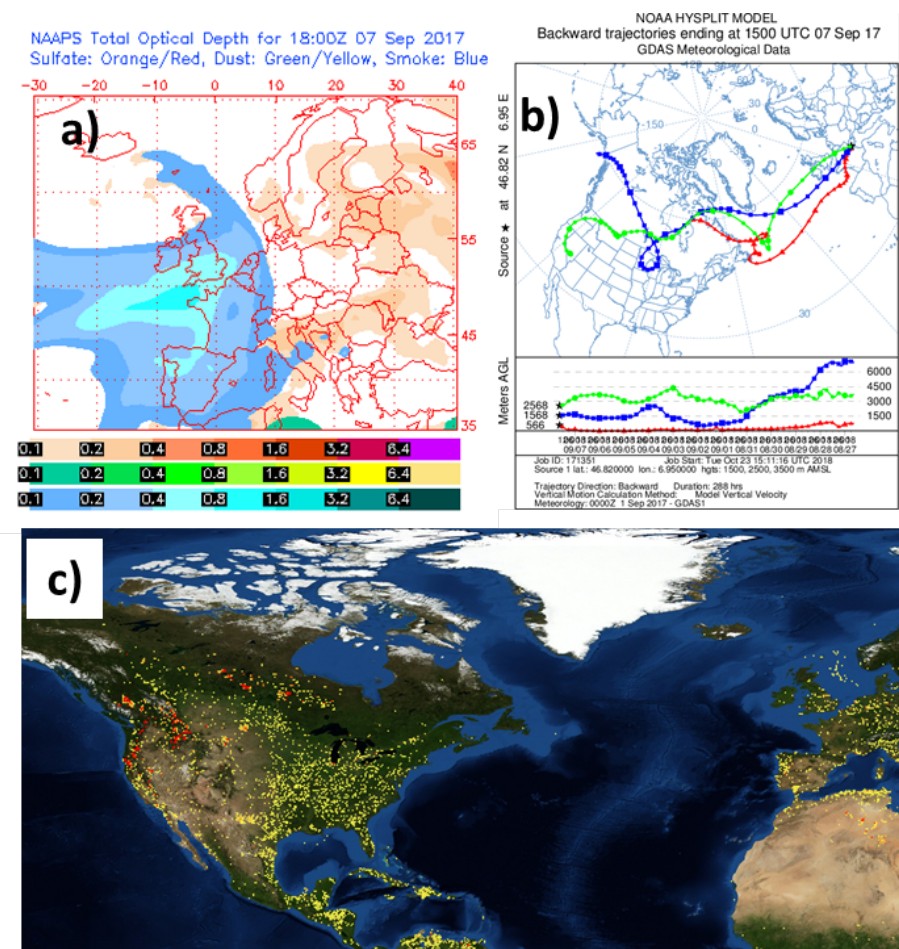

**Figure 4.** a) NAAPS Total Optical Depth for 18 UTC on 7th September 2017. b) Backward trajectories from NOAA Hysplit model ending at 15 UTC 7 Sep 2017 calculated for the altitudes 1500, 2500 and 3500 m asl. c) Fire map from VIIRS instrument for the period 25 August 2017 - 3 September 2017.





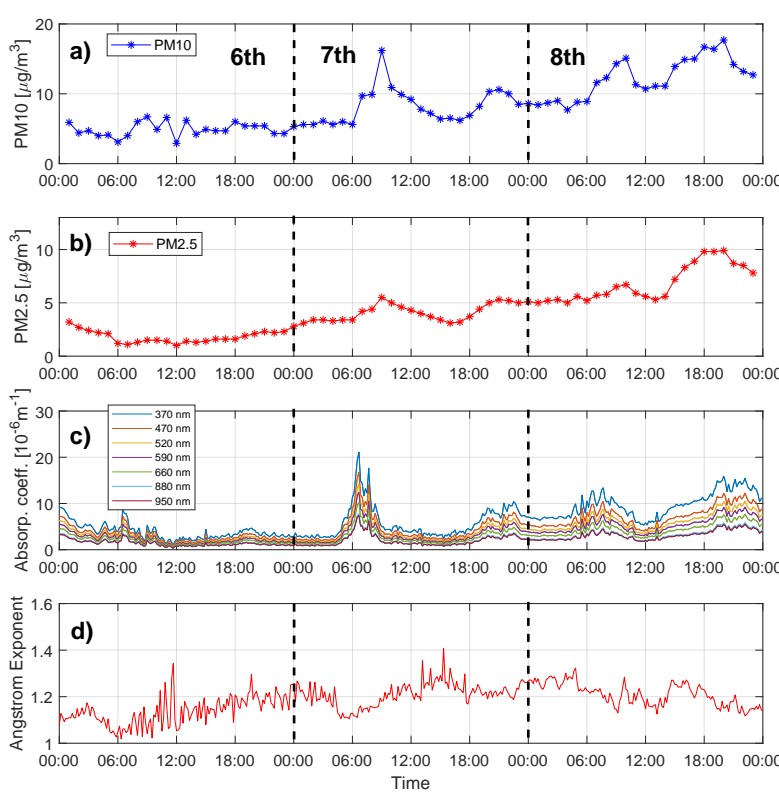

**Figure 5.** Surface PM10 (a) and PM2.5 (b) concentrations at Payerne from 6 to 8 of September 2017. Absorption coefficient at 7 wavelengths (d) and absorption Angstrom exponent (e) are obtained from aethalometer measurements.



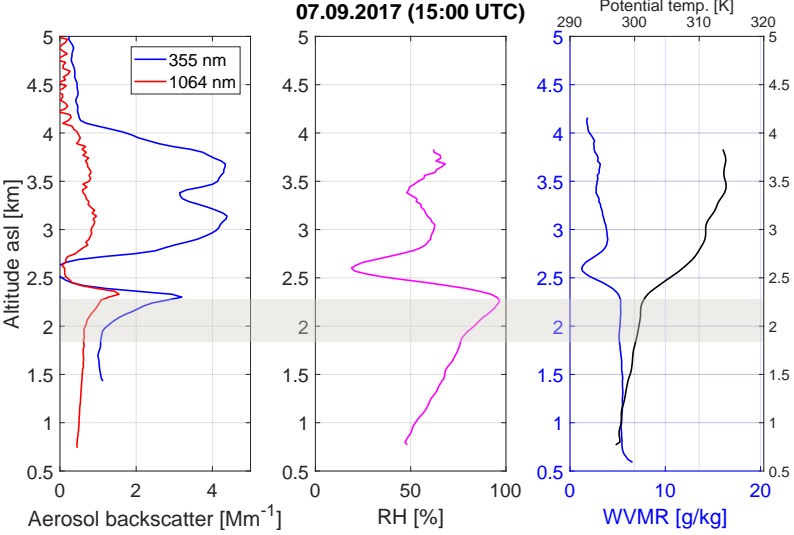

**Figure 6.** Vertical profiles of aerosol backscatter coefficient at 355 and 1064 nm (a), relative humidity (b) and potential and water vapour mixing ratio (c) for the time interval 15:00-15:30 UTC on 7 September 2017.

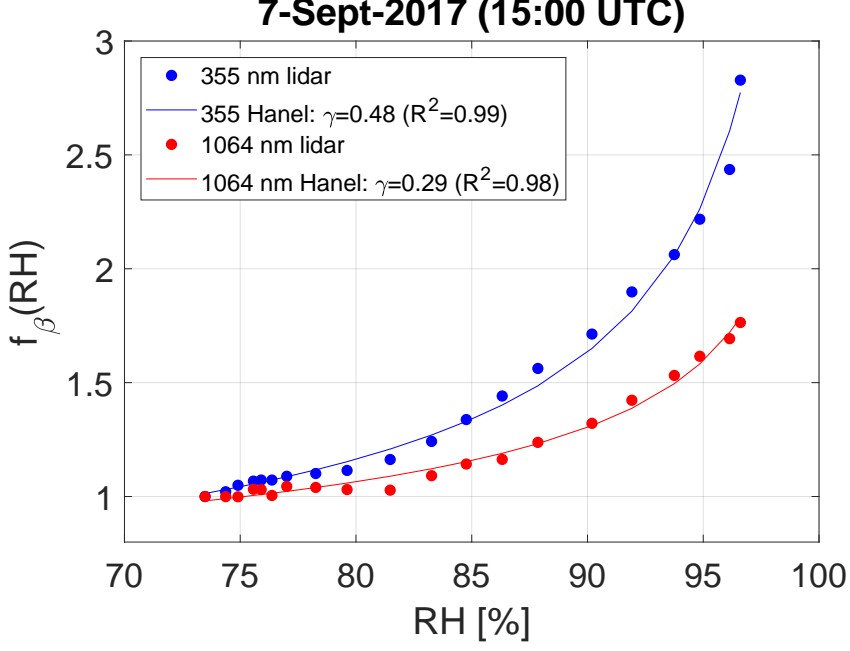

**Figure 7.** Backscatter enhancement factor at 355 and 1064 nm retrieved at 15:00 on 7 September 2017.





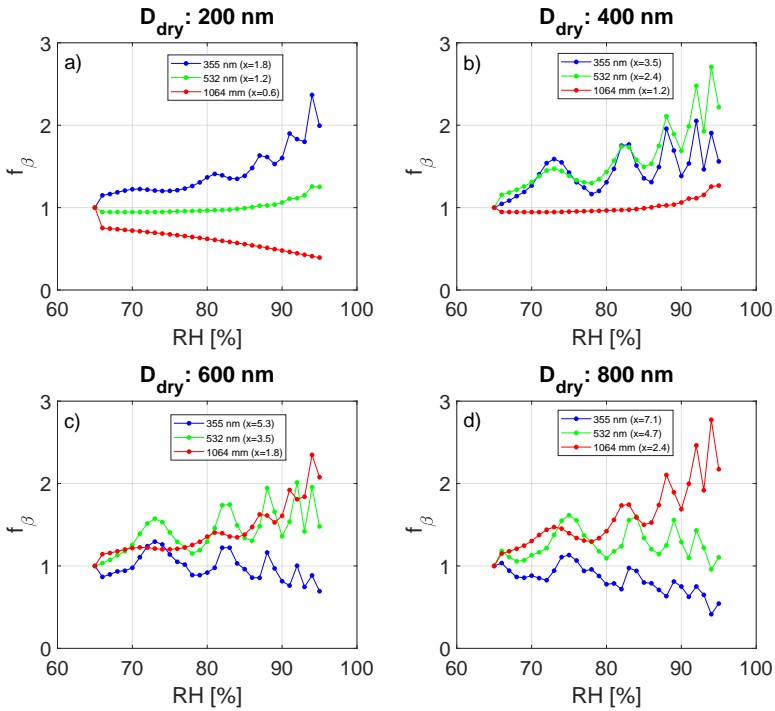

**Figure 8.** Backscatter enhancement factor calculated from Mie simulations at 355, 532 and 1064 nm and for different particle diameters. The size parameter (x) has been indicated for each configuration of wavelength and size particle.

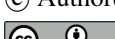


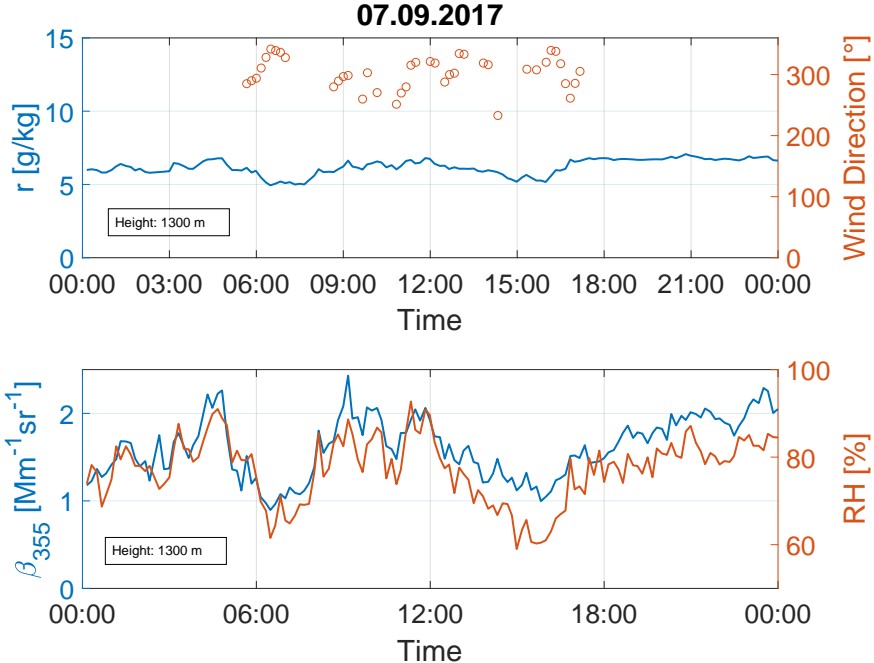

**Figure 9.** Top: Evolution $r$ and wind direction (top) and $\beta_{355}$ and RH (bottom) at 1.3 km (asl) on 07 September 2017.

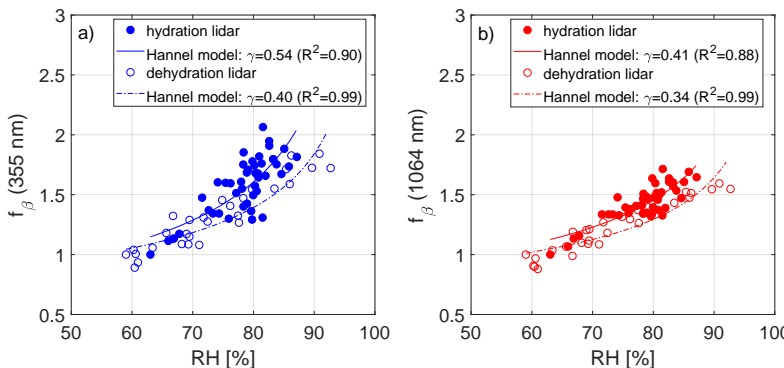

**Figure 10.** Humidograms at 355 nm (a) and 1064 nm (b) retrieved from continuous measurements for hydration and dehydration processes on 7th September 2017.



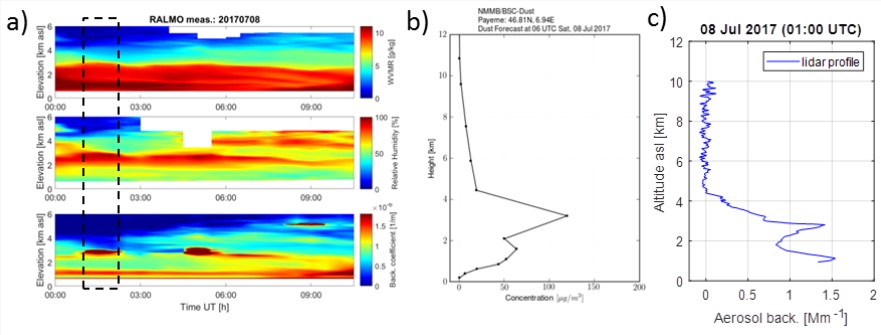

**Figure 11.** a) Quicklook of $r$, RH and backscatter coefficient at 355 nm from Ralmo measurements on 8th July 2018. b) Dust concentration profile from NMMB/BSC model forecast at 06 UTC on 8th July 2017. c) Aerosol backscatter profile at 355 nm from Ralmo lidar at 01 UTC on 8th July 2017.

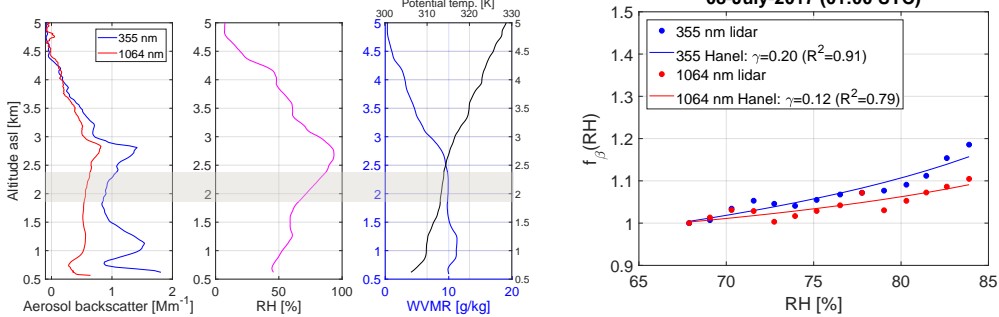

**Figure 12.** Left: Lidar vertical profiles of aerosol backscatter coefficient at 355 and 1064 nm (left panel), RH (central panel), and potential temperature and $r$ (right pannel) obtained between 01:00 and 01:30 on 8th July 2017. Right: Backscatter enhancement factor at 355 and 1064 nm retrieved for that profiles.




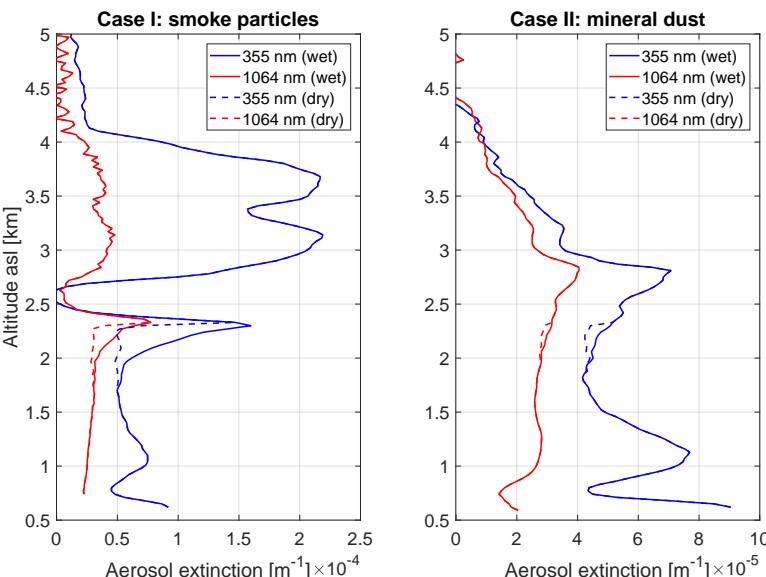

**Figure 13.** Aerosol extinction profiles at 355 and 1064 nm considering wet (ambient) and dry conditions (in the layer with aerosol hygroscopicity) for Case I (left panel) and Case II (right panel).