# Peer review of "Towards continuous monitoring of aerosol hygroscopicity by Raman lidar measurements at the EARLINET station of Payerne"

_Atmospheric Chemistry and Physics, 2019_

## Referee Comment (RC1) · Anonymous Referee #1 · 30 Apr 2019

Summary:

The manuscript describes the retrieval of aerosol hygroscopicity from Raman lidar measurements. Also included is a comparison of mixing ratio, temperature and relative humidity profiles from lidar with those from radiosondes. This is a comprehensive paper of, in my opinion, very high interest to the atmospheric remote sensing community. Therefore, I recommend this work to be published in ACP. However, there are some minor points that need to be addressed before publication. My general comments are given below. Please note that specific comments and technical corrections are provided in the attached document.

[Figure]

General comments:

Most importantly, I'm missing a discussion of uncertainties in the part of this study about the impact of hygroscopicity of aerosols on the radiative budget.

Another aspect to correct is the consistency within this manuscript. Generally it is recommended to use present tense describing established knowledge and previously published work, and for presentation of results (Figure 1 shows . . .), and to use past tense describing methods and results, and for referencing (Author X reported . . .). I have added comments in appropriate places throughout the manuscript, but it would be helpful to give the finished manuscript to a native English speaker to check the language. There are also inconsistencies in some units, especially altitude measures are given in m and km, and in date formats. In figure captions alone, there are many different date formats (8 of September 2017, 7th September 2017, 3 September 2017, 07 September 2017), please homogenise these throughout the text and captions.

Please also note the supplement to this comment:
https://www.atmos-chem-phys-discuss.net/acp-2019-289/acp-2019-289-RC1-supplement.pdf

[Figure]

**Supplement:**

**Review of manuscript with number: acp-2019-289**
Towards continuous monitoring of aerosol hygroscopicity by Raman lidar measurements at the
EARLINET station of Payerne

April 30, 2019

**Summary**
The manuscript describes the retrieval of aerosol hygroscopicity from Raman lidar measurements. Also included is a comparison of mixing ratio, temperature and relative humidity profiles from lidar with those from radiosondes. This is a comprehensive paper of, in my opinion, very high interest to the atmospheric remote sensing community. Therefore, I recommend this work to be published in ACP. However, there are some minor points that need to be addressed before publication.
My comments and corrections are given below.

**General comments**
Most importantly, I'm missing a discussion of uncertainties in the part of this study about the impact of hygroscopicity of aerosols on the radiative budget.

Another aspect to correct is the consistency within this manuscript. Generally it is recommended to use present tense describing established knowledge and previously published work, and for presentation of results (Figure 1 shows …), and to use past tense describing methods and results, and for referencing (Author X reported …). I have added comments in appropriate places throughout the manuscript, but it would be helpful to give the finished manuscript to a native English speaker to check the language. There are also inconsistencies in some units, especially altitude measures are given in m and km, and in date formats. In figure captions alone, there are many different date formats (8 of September 2017, 7th September 2017, 3 September 2017, 07 September 2017), please homogenise these throughout the text and captions.

**Specific comments**

**Page 1**
**Title**
You do not highlight the automatization of the detection of hygroscopicity. Shown are case studies from automatic and continuous measurements, but the hygroscopicity retrieval was done "manually", as far as I understood. You could crop the title to "Aerosol hygroscopicity from Raman lidar measurements at the EARLINET station of Payerne".

**Abstract**
**Line 7**
"whole troposphere" is only valid for nighttime retrievals. Please rephrase.

**Line 11**
"rural aerosol": You only mention a mix of "local" and smoke particles once in the main text, and once in the conclusion. Elsewhere you only refer to smoke. You could update smoke references in the text to "smoky mix" or similar.

**Line 17**
Please see my comment on the use of "significant" in the main text (page 13, line 20).

**Page 8**
**Line 2, Figure 1**
Use either "r" or "mixing ratio", and either "T" or "temperature" in the text. After introducing "r" on page 5, line 12, you keep using "mixing ratio". When discussing the results you sometimes use "r", and sometimes "mixing ratio". It should be consistent throughout. Please update occurrences of "r"/"mixing ratio" in the text.
T is first used on page 6, line 8, but is not introduced as meaning temperature, nor used in other parts of the text.

**Line 31-32**
Am I correct that you averaged the mean bias below 2.1 km asl, where it was positive? In that case, please replace "was +2.0" with "was on average +2.0", and "was -4" with "was on average -4".

**Page 9**
**Line 3**
You are using different altitude ranges to average mean bias (2.1 km asl) and standard deviation (2.0 km asl) of the RH nighttime comparison, and yet different when looking at temperature and RH daytime (5 km asl). The only reason I can see, is that the mean bias of RH changes from positive to negative at 2.1 km asl. Please provide justifications for your choice of altitudes in the other two cases.

**Line 6**
You refer here to "errors". What are the errors/uncertainties of RS and lidar profiles? It is not specified. Or do you mean the difference between the profiles?

**Line 30**
Please explain NAAPS and add reference.

**Line 32**
I don't think that the lowest trajectory (arriving at 566 m agl, red) indicates air mass origin. It is too close to the ground throughout its journey to allow any conclusion on air mass origin.

**Page 10**
**Figure 5**
You show, but do not discuss PM10. Either remove this panel from the figure, or discuss what is shown.

**Line 22**
Why are potential temperature and mixing ratio auxiliary information? Besides, you sometimes use "temperature" and sometimes "potential temperature". Please be clear and specific, when which is used.

**Line 34-35**
Referring to "indicating a lower sensitivity of this wavelength to the aerosol hygroscopic growth": Do you mean this applies to this case? Or generally? Would it depend on the aerosol type?

**Page 11**
**Line 2**
Please rephrase the sentence starting with "This parameter …". I don't understand, especially the last part "observed in the aerosol property".

**Line 6**
What do you mean with "consistent"? The trend is the same (lower at longer wavelength), but the values are different. How large is uncertainty or variability of the Hänel parameter? This information would help to judge, what falls into the "consistent" range.

**Line 21**
You write that backscatter is sensitive to wavelength and particle size, **as expected**. This contradicts somewhat the first sentence of this paragraph. Please explain, or rephrase.

**Line 22**
You write, Mie scattering regime is expected for $x \approx 1$. However, x is as large as 7.1 in figure 8d. Please discuss how representative or applicable your Mie simulations are in such conditions, especially for wavelength 355 nm.

**Page 12**
**Line 10**
Out of curiosity, did you also look at dehydration along the vertical?

**Page 13**
**Line 12**
Did you try this on night-time measurements using directly the Raman lidar extinction profiles? Please briefly discuss here or below, what impact the choice of LR has on AOD, ARE and FE.

**Line 15**
Is AOD in table 1 columnar AOD from the full integrated lidar extinction profiles (which height range?), or just layer AOD? As the only change (according to figure 13) occurs in one layer per case, it would also be useful to see these values for the layers only. Please discuss if it is reasonable, that there is no hygroscopicity in other altitude regions. (See my comment on figure 13)

**Line 17**
You wrote it in the table title. Please also add here a short remark that the relative values are relative to dry values.

**Line 20**
It is hard to judge significance, if no uncertainties are provided. Please specify what you mean with "significant".

**Table 1**
Please provide an estimate of uncertainties, either in the table or in the text.

**Line 31**
Please add a reason, why you chose to optimise solar zenith angle for Case 1 rather than Case 2. I assume it's because the hygroscopic effect was stronger in Case 1, but it is not mentioned.

**Page 14**
**Line 10**
Does this not contradict the previous sentence? It sounds to me, that it would not be necessary. As long as AOD and ARE are right, FE is not sensitive to hygroscopicity. Please elaborate, I cannot follow this statement.

**Page 23**
**Figure 3 (and figure 11 a)**
Please consider using a different colour map, for reasons outlined here among other sources: https://www.mathworks.com/tagteam/81137_92238v00_RainbowColorMap_57312.pdf (see page 3 for a brief overview). I find cubehelix a good replacement (or the reversed version of cubehelix, from light to dark colour).

**Page 26**
**Figure 7**
Did you use instantaneous lidar measurements here, or again the average from 15:00 to 15:30? Please specify.

**Page 27**
**Figure 8**
Is the size parameter x a result of the simulations, or was it an input?

**Page 29**
**Figure 11**
Show plot a) as separate figure.
Increase font size in plot b). Did you create plot 11b? If not, add source. Possibly like: "Adapted from [source]". If you plotted the model output, please keep style consistent with other figures in this manuscript.
Remove legend in plot c), or add meaningful label.
In the caption, replace "Quicklook of r, RH and backscatter coefficient at 355 nm from Ralmo measurements on" with "Same as figure 3, on". Avoid "quicklook", it's too specific to the lidar community.

**Page 30**
**Figure 13**
Those profiles suggest that there was no hygroscopic growth in other regions along the profile. How realistic is that? I think it would be better to, for example, plot the full wet profile as faint line, the wet profile within the studied layer as bold line, and the dry profile within the layer as dashed line, but without connecting it to the wet profile.

**Technical corrections**

**Page 1**
Line 14        replace "is" with "was"
Line 16        be specific about "this type of aerosol"
**Page 2**
Line 4         remove "altering also in this way", and rephrase the following part of this sentence to
               "also altering the global radiative budget (indirect effects)(…) in this way."
Line 9         replace "has been" with "was";
               remove "in order"
Line 10        replace "kind" with "kinds"
Line 12        replace "as" with "like"
Lines 15-16    keep either "still" or "yet", remove one of them
Line 17        replace "RH" with "relative humidity (RH)";
               add comma after "water"

| Line 32 | rephrase sentence: "remote sensing [...] since it", or "remote sensors [...] since they", or "remote sensing techniques [...] since they" |
| Line 33 | it is not clear what "this technique" refers to |

**Page 3**

| Line 3 | replace "RS" with "radiosondes (RS)" |
| Line 10 | please explain what "r" is |
| Line 29 | replace "telescope" with "telescopes" |

**Page 4**

| Line 4 | replace "to" with "into" |
| Line 11 | replace "have been" with "were" |
| Line 13 | "temperature" appears twice |

**Page 5**   no comments

**Page 6**

| Line 23 | replace "the particle" with "particles" |
| Line 28 | replace "defined" with "fixed / constant" |
| Line 29 | replace "are" with "were" |
| Line 30 | replace "need" with "needed" |

**Page 7**

| Line 1 | remove "the"; |
| | add comma after "that" |
| Line 3 | replace "use" with "used"; |
| | replace "the profiles" with "profiles of" |
| Line 4 | replace "are" with "were" |
| Line 6 | replace "have been" with "were" |
| Line 11 | replace "An other" with "Another" |
| Line 12 | remove "have" |
| Line 25 | duplication of "of" |
| Line 30 | add comma after "critical" |

**Page 8**

| Line 1 | replace "RSs" with "RS"; |
| | replace "allows" with "allowed" |
| Line 10 | add comma after "retrievals"; |
| | replace "has been" with "was" |
| Line 12 | replace "has been" with "was"; |
| | replace "treated" with "discussed"; |
| | change format of citation to "(Martucci et al., in preparation)" |
| Line 13 | add comma after "UTC)" |
| Line 19 | add comma after "UTC)" |

**Page 9**

| Line 1 | remove "the" |
| Lines 7-10 | I find this difficult to follow. Please rephrase. Commas are your (and your readers') friends! |
| | For example: "In any case it is important to point out the good accomplishment of Ralmo in retrieving RH information. This can be concluded from this intercomparison, in which Ralmo showed very small biases and standard deviations (below 9%RH), which are indicative of accuracy and precision, respectively, of our measurements." |
| Line 12 | add comma after "studies" |
| Line 15 | add comma after "2017"; |
| | rephrase the rest of the sentence, for example: "...with smoke particles present during one, and mineral dust during the other." |
| Line 16 | replace "in" with "of" |

| | |
|---|---|
| Line 18 | replace "pannel" with "panel" |
| Line 19 | replace "planetary" with "atmospheric" |
| Line 20 | replace "PBL" with "ABL", and change throughout the manuscript |
| Line 21 | replace "pattern" with "development"; |
| | replace "along" with "during" |
| Line 24 | replace "is" with "was"; |
| | replace "pannel" with "panel" |
| Line 25 | replace "for" with "during" or "in" |
| Line 31 | add "(blue colour map)" after "concentrations"; |
| | replace "over" with "at" (you refer to surface concentration) |
| Line 32 | replace "from ground to 3 km asl" with "from 1 to 3 km asl" |
| **Page 10** | |
| Line 2 | replace "for" with "in" |
| Line 10 | remove "burning", this is already covered by "combustion" |
| Line 12 | Please specify "that days" (or rather "those days"). Do you mean "in the period from 6th to 8th September"? |
| Line 14 | replace "concentrations" with "concentration"; |
| | replace "were" with "was"; |
| | remove "than" |
| Line 15 | replace "took place" with "occurred" |
| Line 16 | add comma after (AAE) |
| Line 20 | replace "of" with "over" |
| Line 24 | replace "is" with "was" |
| Lines 24-25 | remove either "Simultaneously to this increase," or "for the same layer"; |
| | remove "that there is also"; |
| | replace "can observe" with "observed"; |
| | replace "moving" with "increasing" |
| Line 28 | add comma after "respectively)"; |
| | replace "is" with "was" |
| Line 30 | replace "have been" with "were" |
| Line 31 | add comma after "RH"; |
| | remove "what is" |
| Line 34 | replace "respect its" with "with respect to its" |
| **Page 11** | |
| Line 1 | replace "took a value of" with "was" |
| Line 11 | replace "have been" with "was" |
| Line 13 | remove "have" |
| Line 21 | replace "how" with "that"; |
| | remove 2nd "the"; |
| | replace 2nd "and" with comma |
| Line 22 | add: "where D is particle diameter" after the equation |
| Line 29 | remove "totally" |
| Line 32 | add comma after "cases"; |
| | remove "also means that we" |
| Line 33 | replace "the theory of Mie" with "Mie theory"; |
| | replace "," with "."; |
| | replace "so" with "Hence," (start new sentence) |
| **Page 12** | |
| Line 2 | replace "For" with "In"; |
| | replace "we can observe that r (Fig.9, top) is quite constant (…)" with "we can observe that r was quite constant (…, Fig. 9, top)" |

| | |
|---|---|
| Line 4 | remove "The" |
| Line 6 | remove "again" |
| Line 9 | remove "very" |
| Line 10 | replace "suffered by this aerosol" with "that occurred within this aerosol layer" |
| Line 12 | remove "very"; |
| | remove "either" |
| Lines 14-15 | it sounds like you used a hygroscopic parameter of 0.4 for the later period (16:00 to 23:30); please rephrase; |
| | remove "also" in line 15 |
| Line 21 | replace "pannel" with "panel" |
| Line 22 | explain acronyms "NMMB" and "BSC" |
| Line 23 | add comma after "Europe" |
| Line 24 | replace "in" with "for" or "above" |
| Line 27 | replace "in our station" with "at our station" |
| Line 29 | replace "along" with "throughout" |
| Line 30 | replace "along" with "during" or "throughout" |
| Line 31 | remove parentheses and include as full sentence |
| Line 32 | replace "is" with "was"; |
| | add comma after "measurements"; |
| | replace "analyze" with "analyzed" |
| Line 33 | replace "observe" with "observed" |

**Page 13**

| | |
|---|---|
| Lines 5-6 | put parenthesis and dot on same line as 0.24 (remove space?) |
| Lines 6-7 | rephrase, for example: |
| | As in case 1, we found the opposite spectral dependency compared to Lv et al. (2017). However, we considered a wider spectral range. |
| Line 10 | remove "also"; |
| | remove "the" |
| Line 11 | remove "the", twice in front of "AOD" |
| Line 13 | remove first "the" |
| Line 18 | replace "16.6" with "15.6" (check if table 1 or text is correct); |
| | remove first "the" |
| Line 23 | remove "the" |
| Line 24 | remove first occurrence of "model" |
| Line 28 | add "and relative" after "absolute" |
| Line 31 | remove "the one of" |

**Page 14**

| | |
|---|---|
| Line 13 | replace "monitor" with "observe" or "detect"; "monitor" sounds more like an automated process |
| Line 14 | replace "in almost a continuous way" with "almost continuously" |
| Line 16 | remove "the"; |
| | replace "the particle" with "particles" |
| Line 17 | add comma after "growth"; |
| | replace "have been" with "were" |
| Line 21 | replace "along" with "throughout" |
| Line 27 | replace "in the full" with "throughout the" |
| Line 33 | swap order of sentence: "…in the lower troposphere (…) were very similar to the ones obtained during nighttime." |

**Page 15**

| | |
|---|---|
| Line 14 | add: "as well as the study of Haarig et al. (2017)" |
| Line 20 | remove "very" |

| Line 21 | add comma after "2017)"; |
| | add comma after "particles" |
| Lines 26-27 | remove first "the"; |
| | remove "It"; |
| | move "was also remarkable" to the end of the sentence |
| Line 28 | replace "has been" with "was" |
| Line 30 | remove first "the" |
| Line 31 | add "and relative" after "absolute" |
| Line 32 | please see specific comment for page 13, line 20 |
| Line 33 | replace "was not" with "not having been"; |
| | add comma after "Therefore" |
| Line 34 | replace "have" with "has" |

**Page 16**

| Line 6 | replace "AA" with "AH"? |

**Page 21**

| Table 1 | replace "layer width" with "layer depth" |

**Page 22**

| Figure 1 | start caption with: "Mixing ratio (r), temperature (T) and relative humidity (RH) [...]" |
| Figure 2 | add date after "UTC" |

**Page 23**

| Figure 3 | replace "of" with "of vertical profiles of"; |
| | In the caption "r" is called "water vapour mixing ratio", and in the figure "WVPR". |
| | Please amend figure and caption to "mixing ratio" and "r" to keep it consistent |
| | throughout the manuscript and to avoid confusion. |

**Page 24**

| Figure 4 | replace "Total Optical Depth" with "total optical depth from sulfate (orange/red scale), |
| | dust (green/yellow scale), and smoke (blue scale)"; |
| | possibly replace "1500, 2500, 3500 m asl" with "1000, 2000, 3000 m asl"; Payerne |
| | being roughly at 500 m asl. 3500 m asl is not consistent with the text either (page 9, |
| | line 32). |

**Page 26**

| Figure 6 | explain in caption what the grey shaded area means; |
| | add "temperature" after "potential" |

**Page 27**

| Figure 8 | increase font size in legends |

**Page 28**

| Figure 9 | add "of" after "Evolution" |
| Figure 10 | replace "Hannel" with "Hänel" in figure legends |

**Page 29**

| Figure 12 | replace "pannel" with "panel" (3 times in this caption); |
| | increase size of the three panels on the left; font sizes in left and right plot should be |
| | the same |

---

## Referee Comment (RC2) · Anonymous Referee #2 · 7 May 2019

Review for manuscript "Towards continuous monitoring of aerosol hygroscopicity by Raman lidar measurements at the EARLINET station of Payerne"

Authors study important problem of the aerosol hygroscopic growth basing on long term multiwavelength lidar observations. The research is done on high scientific level. Authors well understand all the issues, when the information about humidification process is extracted from lidar measurements. Manuscript is well written and can be published after minor revisions. The Reviewer 1 provided very detailed review, so I can add just several technical comments.

Title. I agree with Reviewer 1 that title can be shortened.

[Figure]

p.10, ln 24. "From this figure, a marked increase of with altitude is observed for the altitude range between 1.7 and 2.3 km (asl)." The same time for range 1.5-2.0 backscattering doesn't increase significantly, though RH rises. Any ideas why (potential temperature and mixing ratio are quite stable)?

Fig.7. What will happen with these curves and Hanel parameters if starting height is 1.5 km? How sensitive are results to the choice of height interval?

Fig.11-13. Figures should be done in the same style: size, format, fonts, grids should be kept the same. Some fonts are very small, difficult to read. Probably Fig11b,c can be shown on the same plot.

---

## Author Comment (AC1) · 23 Jul 2019

**Authors' response**

We thank the anonymous reviewer for his/her very detailed revision that has helped to improve the quality of the manuscript. According to the referee's reports, the following changes have been performed on the original manuscript and a point-by-point response is included below, where blue colour is related with answers.

**Review of manuscript with number: acp-2019-289**

**Towards continuous monitoring of aerosol hygroscopicity by Raman lidar measurements at the EARLINET station of Payerne**

**Reviewer 1**

**Summary**

The manuscript describes the retrieval of aerosol hygroscopicity from Raman lidar measurements. Also included is a comparison of mixing ratio, temperature and relative humidity profiles from lidar with those from radiosondes. This is a comprehensive paper of, in my opinion, very high interest to the atmospheric remote sensing community. Therefore, I recommend this work to be published in ACP. However, there are some minor points that need to be addressed before publication. My comments and corrections are given below.

**General comments**

Most importantly, I'm missing a discussion of uncertainties in the part of this study about the impact of hygroscopicity of aerosols on the radiative budget.

Another aspect to correct is the consistency within this manuscript. Generally it is recommended to use present tense describing established knowledge and previously published work, and for presentation of results (Figure 1 shows …), and to use past tense describing methods and results, and for referencing (Author X reported …). I have added comments in appropriate places throughout the manuscript, but it would be helpful to give the finished manuscript to a native English speaker to check the language. There are also inconsistencies in some units, especially altitude measures are given in m and km, and in date formats. In figure captions alone, there are many different date formats (8 of September 2017, 7th September 2017, 3 September 2017, 07 September 2017), please homogenise these throughout the text and captions.

In the revised manuscript we have included a discussion about the uncertainties of the radiative transfer model. In addition we have made a careful review of the text in the manuscript (English, date formats, figures …). Below we have answered point by point to all the specific comments.

**Specific comments**

**Title**

You do not highlight the automatization of the detection of hygroscopicity. Shown are case studies from automatic and continuous measurements, but the hygroscopicity retrieval was done "manually", as far as I understood. You could crop the title to "Aerosol hygroscopicity from Raman lidar measurements at the EARLINET station of Payerne".

We agree with the reviewer that we did not discuss about the automatization of aerosol hygroscopic detection in the manuscript. Therefore, following his suggestion we have modified the title of our work in order to give a better description of our study. The new title reads as:

**"Characterization of aerosol hygroscopicity using continuous Raman lidar measurements at the EARLINET station of Payerne"**

Abstract

**Line 7**

"whole troposphere" is only valid for nighttime retrievals. Please rephrase.

The sentence has been rephrased and now reads as (page 1, lines 6-9):

**"A total of 172 RS profiles were used in this intercomparison which revealed a bias smaller than 4%RH and a standard deviation smaller than 10%RH between both techniques in the whole (in lower) troposphere at nightime (at daytime), indicating the good performance of the lidar for characterizing RH."**

**Line 11**

"rural aerosol": You only mention a mix of "local" and smoke particles once in the main text, and once in the conclusion. Elsewhere you only refer to smoke. You could update smoke references in the text to "smoky mix" or similar.

Following the suggestion of the referee we have used the expression "smoke mixture" in the reviewed manuscript to refer to the mix of smoke and local aerosol.

**Line 17**

Please see my comment on the use of "significant" in the main text (page 13, line 20).

The uncertainties associated to the radiative transfer model (GAME) have been discussed in Section 6 along with the validity of our results. Based on that, we also consider more appropriated remove the word "significant" in this sentence. It reads now as (page 1, lines 18-19):

**"The model showed an impact with an increase in absolute value of 2.4 W/m$^2$ at the surface with respect to the dry conditions for the hygroscopic layer of Case I (smoke mixture). "**

**Page 8**

**Line 2, Figure 1**

Use either "r" or "mixing ratio", and either "T" or "temperature" in the text. After introducing "r" on page 5, line 12, you keep using "mixing ratio". When discussing the results you sometimes use "r", and sometimes "mixing ratio". It should be consistent throughout. Please update occurrences of "r"/"mixing ratio" in the text. T is first used on page 6, line 8, but is not introduced as meaning temperature, nor used in other parts of the text.

We have corrected this inconsistency and "r" has been used instead of "mixing ratio" through the text after it was introduced the first time.

The same has been done for the next physical parameters that have been defined and then their corresponding symbols have been used:

- Aerosol backscatter coefficient → $\beta^{aer}$ ;
- Temperature → T ;
- Relative humidity → RH ;
- Aerosol backscatter enhancement factor → $f_\beta$ ;
- Potential temperature → $\theta$ ;

**Line 31-32**

Am I correct that you averaged the mean bias below 2.1 km asl, where it was positive? In that case, please replace "was +2.0" with "was on average +2.0", and "was -4" with "was on average -4".

When we talk about the bias profile we refer to the mean RH deviation profile. So in this sentence with "the mean bias in the lower part" we refer to the arithmetic average of the profile in its lower part. The standard deviation of the values used for the average is provided but not explained. So, in order to clarify this point we have rephrased the sentence in the following way (page 8, lines 22-23):

 **"The mean bias and standard deviation along the region from ground to 2.1 km asl is +2.0±0.9 %RH while it increases to -4±2 %RH in the range 2.1-9 km".**

**Page 9**

**Line 3**

You are using different altitude ranges to average mean bias (2.1 km asl) and standard deviation (2.0 km asl) of the RH nighttime comparison, and yet different when looking at temperature and RH daytime (5 km asl). The only reason I can see, is that the mean bias of RH changes from positive to negative at 2.1 km asl. Please provide justifications for your choice of altitudes in the other two cases.

As the referee pointed out, the main reason to give different altitude ranges for the RH statistics compared to the T statistics is to quantify the different behaviour that is observed for the RH bias in those ranges. But we agree that is confusing to give different ranges for the RH bias and standard deviation. We have therefore modified the discussion in the manuscript and now we refer to the same altitude range for the bias and std discussion. The text reads now as (page 8, lines 29-30):

**"The mean *RH* standard deviation in the lower troposphere (from ground to 2.1 km) was 6.5±1.3%RH while it was 8.5±1.5%RH above this altitude."**

**Line 6**

You refer here to "errors". What are the errors/uncertainties of RS and lidar profiles? It is not specified. Or do you mean the difference between the profiles?

We agree that in the sentence it is not clear what kind of errors we are referring to. In order to clarify it we have rephrased it (page 8, lines 33-34):

**"Above 5 km (asl) and during daytime the SNR of RALMO is smaller compared to night-time measurements due to the solar background, for this reason the RH profiles were not calculated above this altitude in order to avoid large uncertainties."**

**Line 30**

Please explain NAAPS and add reference.

Following the referee's suggestion, the meaning of the acronym and a reference has been included in the sentence. It reads now as (page 9, lines 20-21):

**"According to the NAAPS (Navy Aerosol Analysis and Prediction System) (Christensen, 1997) model, 7th September 2017 at 18:00 UTC is characterized by the presence of smoke above the measurement station (Fig. 4a).**

**Line 32**

I don't think that the lowest trajectory (arriving at 566 m agl, red) indicates air mass origin. It is too close to the ground throughout its journey to allow any conclusion on air mass origin.

We agree with the referee that the lowest trajectory could be more problematic. Since we are more interested in the air masses arriving at 1.8-2.3 km, where the hygroscopic layer is located, we have removed that trajectory. The sentence in the manuscript referring to this figure has been slightly modified. It reads now as (page 9, lines 22-23):

**"The Hysplit backtrajectories analysis indicated that the air masses above our station in the lower layers of the troposphere had their origin in North America (Fig. 4b)."**

**Page 10**

**Figure 5**

You show, but do not discuss PM10. Either remove this panel from the figure, or discuss what is shown.

Since the plot of the PM10 concentration is not discussed in the manuscript and does not add relevant information to our discussion we have removed this panel from Fig. 5. The caption of the figure has also been modified accordingly.

**Line 22**

Why are potential temperature and mixing ratio auxiliary information? Besides, you sometimes use "temperature" and sometimes "potential temperature". Please be clear and specific, when which is used.

We call "auxiliary information" to the profiles of potential temperature and water vapour mixing ratio because although these parameters are not used explicitly to assess the aerosol hygroscopicity they are used to guarantee good mixing conditions within the analysed layer. This is one of the requirements for the vertical analysis of aerosol hygroscopicity as it is explained in the methodology (subsection 3.2).

Then in the manuscript, when we refer to "temperature" we mean physical temperature at a determined pressure level while when we refer to "potential temperature" we mean the temperature that a parcel of fluid at pressure(P) would attain if adiabatically brought to a standard reference pressure of 1000 millibars. Potential temperature is only used in this study to assess the aerosol mixing conditions as it was indicated before.

**Line 34-35**

Referring to "indicating a lower sensitivity of this wavelength to the aerosol hygroscopic growth": Do you mean this applies to this case? Or generally? Would it depend on the aerosol type?

Yes, we refer to this case (smoke mixture) in this sentence, but it was also observed in the other case (with presence of mineral dust) although in a weaker way. Opposite spectral behaviours have been observed in other studies (higher sensitivity at longer wavelengths). The differences found in this work respect to previous ones have been extensively discussed in the manuscript and are attributed to the different particle's sizes found in this study. The results have been explained using Mie's theory and it is one of the important conclusions of our paper.

**Page 11**

**Line 2**

Please rephrase the sentence starting with "This parameter …". I don't understand, especially the last part "observed in the aerosol property".

We agree with the referee that the sentence is unclear. We have rephrased the sentence in the following way (page 10, lines 26-27):

 **"This parameter is proportional to the aerosol hygroscopicity and it had values of 0.48±0.08 at 355 nm and 0.29±0.08 at 1064 nm."**

**Line 6**

What do you mean with "consistent"? The trend is the same (lower at longer wavelength), but the values are different. How large is uncertainty or variability of the Hänel parameter? This information would help to judge, what falls into the "consistent" range.

The uncertainty in the Hänel parameter has been calculated for each individual fitting line. The fitting was performed including the errors in the aerosol backscatter coefficient (10%, Ansmann et al., 2002) and in RH (7%RH, based on our RH statistical analysis). This new information (γ uncertainties) has been included in the different figures and also in the text. It reads now as (page 10, lines: 30-33):

**"Although the *RH* range observed for this time interval (70-93%) was slightly different from the one showed in Fig. 6, showed consistent results, within the associated uncertainties, for both time intervals ($\gamma_{355}$=0.57±0.14 and $\gamma_{1064}$=0.35±0.14). "**

**Line 21**

You write that backscatter is sensitive to wavelength and particle size, as expected. This contradicts somewhat the first sentence of this paragraph. Please explain, or rephrase.

We don't see any contradiction in this point. With the first sentence of the paragraph "We performed Mie simulations in order to understand if the spectral dependency observed in our study could be realistic or not" we are indicating that we used Mie simulations to explain the observed spectral dependency (with different behaviour than in other studies). But it does not mean that was not clear that the backscatter was sensitive to the wavelength and particle size.

**Line 22**

You write, Mie scattering regime is expected for x≈1. However, x is as large as 7.1 in figure 8d. Please discuss how representative or applicable your Mie simulations are in such conditions, especially for wavelength 355 nm.

What we mean with x≈1 is that for Mie scattering the particle are about the same size as the wavelength, unlike Rayleigh scattering (x<<1: small particle compared with the wavelength) and Geometric scattering (x>>1: large particle compared with the wavelength). Strictly speaking we can say that we are in Mie scattering domain when x is between 0.2 and 2000. Therefore the simulated results presented in our study are all representative of the Mie scattering and the problematic part could come from the fact that we are departing from the sphericity assumption for big particles. This point was pointed out in the manuscript.

**Page 12**

**Line 10**

Out of curiosity, did you also look at dehydration along the vertical?

In the two presented cases of this study we did not observe dehydration processes in the vertical. However we expect to find cases with these conditions in our long-data set of lidar measurements. A statistical analysis of aerosol hygroscopic properties is ongoing and the study will be submitted for publication and the end of the year.

**Line 12**

Did you try this on night-time measurements using directly the Raman lidar extinction profiles? Please briefly discuss here or below, what impact the choice of LR has on AOD, ARE and FE.

Extinction profiles were not available for these two cases because it was not possible to characterize correctly the overlap functions (function that characterizes the incomplete overlap between the laser beam and the field of view of the telescope). This is the reason why our study is focused on the effect of hygroscopicity on aerosol backscatter coefficient which is not affected by this incomplete overlap issue (it is calculated as the ratio of two signals affected by the same incomplete overlap and therefore it cancels out).

Regarding the impact of LR we know that by definition that is proportional to the AOD, and it should also produce a proportional variation of the ARE. In the next table we can confirm these results for two extreme values of LR (10 and 90 sr). The variation in AOD and absolute values of the ARE is large for these two values of LR, but the relative values remain almost constant. The absolute values are affected by large uncertainty because of the selection of the LR, but the relative contribution due to hygroscopicity remains almost constant in both cases. Anyway, LR values of 90 and 10 are too extreme and the LR of 50 sr is more realistic based on literature and the aerosol type we are analysing, so the uncertainty should be much lower.

| | $LR(sr)$ | $AOD$ $(480nm)$ | $AOD_{dry}$ $(480nm)$ | $\Delta AOD$ | $ARE$ $(W \cdot m^{-2})$ | $ARE_{dry}$ $(W \cdot m^{-2})$ | $\Delta ARE$ |
|---|---|---|---|---|---|---|---|
| *Case I* | 90 | 0.52 | 0.49 | 0.03 | -78.39 | -74.54 | -3.85 (5.2%) |
| | 10 | 0.06 | 0.05 | 0.01 | -9.52 | -9.01 | -0.51 (5.6%) |
| | $LR(sr)$ | $AOD$ $(480nm)$ | $AOD_{dry}$ $(480nm)$ | $\Delta AOD$ | $ARE$ $(W \cdot m^{-2})$ | $ARE_{dry}$ $(W \cdot m^{-2})$ | $\Delta ARE$ |
| *Case II* | 90 | 0.272 | 0.270 | 0.002 | -49.08 | -48.82 | -0.26 (0.5%) |
| | 10 | 0.030 | 0.030 | 0.000 | -5.46 | -5.44 | -0.02 (0.3%) |

We have added a comment in Section 6 regarding the effect that the selection LR could have on AOD. It read as (page 13, lines: 5-6):

"**Although the choice of LR could affect to the AOD, the relative contribution due to aerosol hygroscopicity would remain almost constant.**

**Line 15**

Is AOD in table 1 columnar AOD from the full integrated lidar extinction profiles (which height range?), or just layer AOD? As the only change (according to figure 13) occurs in one layer per case, it would also be useful to see these values for the layers only. Please discuss if it is reasonable, that there is no hygroscopicity in other altitude regions. (See my comment on figure 13)

The AOD is the column AOD calculated from the lowest to the highest point of the retrieved vertical profile. The PBL AOD is given only for Case I, and the values are reported with an asteric (*), see Table 1. The objective of Section 6 is to quantify the effect of the hygroscopic growth in a lofted

aerosol layer on the ARE at the surface. For this, the knowledge of the extinction coefficient in the whole column is necessary (not only in the hygroscopic layer). It is surely interesting to see what would be the hygroscopic effect locally in the hygroscopic layer (this could be done by plotting the profiles of the heating/cooling rates) but the authors believe this is out of the scope of the paper.

Small modifications have been done in Table 1:

- Caption: indication that we refer to column AOD.
- Columns about FE have been deleted (they did not add any valuable information, see answer to referee's comment (page 14, line 10))
- Some numbers were mixed between rows, now they are correct.

**Line 17**

You wrote it in the table title. Please also add here a short remark that the relative values are relative to dry values.

Following the referee's suggestion we have indicated this observation in this part of the manuscript text (page 13, lines: 10-11):

"**For the two cases analyzed here, the increase in AOD at 355 nm related to the hygroscopic growth in the analyzed layers is ΔAOD = 0.017 (with ΔAOD = AOD - AOD$_{dry}$) for Case I and ΔAOD = 0.001 for Case II.**"

**Line 20**

It is hard to judge significance, if no uncertainties are provided. Please specify what you mean with "significant".

In this sentence the authors want only to point out in a qualitative way that an increase of 16% in the PBL AOD due to hygroscopic growth is considerable and much larger than the errors associated to the calculation of this parameter. We found the adjective "significant" appropriated to remark this point.

**Table 1**

Please provide an estimate of uncertainties, either in the table or in the text.

In this part of the study, we are mostly interested in studying the relative differences observed in the aerosol radiative forcing when using the "dry" and the "wet" profiles. It is worthy to note that the SSA, asymmetry factor and the meteorological and surface properties are assumed to remain constant during the humidification process and only variations in the AOD are considered. Because of the thin layers affected by hygroscopic growth and the relatively low γ values obtained in the analyzed cases, differences between dry and wet AOD are quite small and an evaluation of the uncertainty is not straightforward. The uncertainty introduced by assuming the SSA and asymmetry factor constant is expected to be negligible according to results obtained in the sensitivity tests performed in Granados-Muñoz et al. (2019), where it was observed that the largest uncertainties for the ARE were associated to uncertainties in the input AOD.

The next lines have been added in Section 6 to clarify the previous points and indicate the uncertainties found for this model in Granados-Muñoz et al. (2019). The text reads as (page 13, lines: 25-30):

"**The interpretation of these results needs to be done carefully. Even though radiative transfer models do not provide an estimation of the ARE uncertainties, a sensitivity test performed in Granados-Muñoz et al. (2019) showed that an uncertainty in the AOD of 0.05 can lead to uncertainties in the ARE of up to 30%. The values of ΔARE obtained here are certainly within this uncertainty limits and an accurate quantitative estimation of the hygroscopicity contribution to the ARE is quite complex. However it is necessary to highlight that our focus here is on the relative contribution of hygroscopic aerosol to the ARE when compared to dry conditions.**"

**Line 31**

Please add a reason, why you chose to optimise solar zenith angle for Case 1 rather than Case 2. I assume it's because the hygroscopic effect was stronger in Case 1, but it is not mentioned.

As the referee indicates the stronger hygroscopicity observed in case I was one of the reasons to choose that solar zenith angle, but another was that case 2 was during nighttime (01UTC), so without solar irradiance, therefore ARE computations were not feasible. Both reasons have been mentioned in the text (page 13, lines: 22-23):

"**The calculation was done for a solar zenith of 30° corresponding to Case I, which presented stronger hygroscopicity and occurred at daytime (15:00 UTC).**"

**Page 14**

**Line 10**

Does this not contradict the previous sentence? It sounds to me, that it would not be necessary. As long as AOD and ARE are right, FE is not sensitive to hygroscopicity. Please elaborate, I cannot follow this statement.

Yes, we wanted to point out that aerosol hygroscopicity is important for ARE but not for FE. In our case, the insensitivity of FE was expected since we only change AOD in our simulations. However, since the discussion about FE does not add any useful information to our study and it could lead to some confusion we have decided to delete from the paper/table 1 all comments related to FE.

So the last part of the paragraph reads now as:

"**In Case I, although the hygroscopic growth affects only a thin layer (only 600 m width) and the γ values are relatively low (0.48), the aerosol hygroscopic growth effect on the ARE is still quite noticeable. These results point out that in more favourable conditions, namely thicker layers where the hygroscopic effect occur and particles with stronger hygroscopic properties, the aerosol hygroscopic effect on the optical and radiative properties could be quite considerable. Therefore, we can conclude that including aerosol hygroscopic properties in climate model calculations is key for improving the accuracy of aerosol forcing estimates.**"

**Page 23**

**Figure 3 (and figure 11 a)**

Please consider using a different colour map, for reasons outlined here among other sources: https://www.mathworks.com/tagteam/81137_92238v00_RainbowColorMap_57312.pdf (see page 3 for a brief overview). I find cubehelix a good replacement (or the reversed version of cubehelix, from light to dark colour).

Following the suggestion of the referee we have remade both figures using the cubehelix colour map.

**Page 26**

**Figure 7**

Did you use instantaneous lidar measurements here, or again the average from 15:00 to 15:30? Please specify.

The backscatter enhancement factors were calculated from the averaged signals from 15:00 to 15:30. It has been indicated in the caption of the figure. It reads now as:

"$f_\beta$ at 355 and 1064 nm retrieved from the lidar profiles (layer: 1.7-2.3 km asl) at the time interval from 15:00-15:30 UTC on 7th September 2017."

**Page 27**

**Figure 8**

Is the size parameter x a result of the simulations, or was it an input?

The size parameter (x) is an input of the simulations which is a function of the particle size and wavelength. It is indicated in the manuscript (page 11, line 23): "Figure 8 shows the $f_\beta$ from the Mie calculation as a function of RH for different wavelengths and particle sizes."

**Page 29**

**Figure 11**

Show plot a) as separate figure.

Increase font size in plot b). Did you create plot 11b? If not, add source. Possibly like: "Adapted from [source]". If you plotted the model output, please keep style consistent with other figures in this manuscript.

Remove legend in plot c), or add meaningful label.

In the caption, replace "Quicklook of r, RH and backscatter coefficient at 355 nm from Ralmo measurements on" with "Same as figure 3, on". Avoid "quicklook", it's too specific to the lidar community.

All the changes suggested by the referee have been implemented. The figure has been separated in two (Fig. 11 and 12) and the text in the manuscript which refers to the figures properly modified.

**Page 30**

**Figure 13**

Those profiles suggest that there was no hygroscopic growth in other regions along the profile. How realistic is that? I think it would be better to, for example, plot the full wet profile as faint line, the wet profile within the studied layer as bold line, and the dry profile within the layer as dashed line, but without connecting it to the wet profile.

We consider that having hygroscopic growth only in a layer can be a very realistic situation since in the rest of the profile you could not reach hygroscopic conditions (high RH). In this section of the paper we just want to evaluate what is the effect of this layer, in which we can guarantee that there was hygroscopic growth, on the Earth's radiative balance.

Regarding Fig. 13 (now Fig. 14), we have plotted the different profiles according to the referee's suggestions.

**Technical corrections**

**Page 1**

Line 14 replace "is" with "was" → done

Line 16 be specific about "this type of aerosol" → Done. " …. dehydration of the smoke mixture."

**Page 2**

Line 4 remove "altering also in this way", and rephrase the following part of this sentence to

"also altering the global radiative budget (indirect effects)(…) in this way." → done

Line 9 replace "has been" with "was"; → done

remove "in order" → done

Line 10 replace "kind" with "kinds" → done

Line 12 replace "as" with "like" → done

Lines 15-16 keep either "still" or "yet", remove one of them → done

Line 17 replace "RH" with "relative humidity (RH)"; → RH was already introduced in the abstract

add comma after "water" → done

Line 32 rephrase sentence: "remote sensing [...] since it", or "remote sensors [...] since they",

or "remote sensing techniques [...] since they" → We have rephrased it as: "remote sensing techniques [….] since they …..

Line 33 it is not clear what "this technique" refers to → We refer to remote sensing techniques. This point has been clarified.

**Page 3**

Line 3 replace "RS" with "radiosondes (RS)" → It was already introduced in the abstract

Line 10 please explain what "r" is → It was introduced in the abstract

Line 29 replace "telescope" with "telescopes" → done

**Page 4**

Line 4 replace "to" with "into" → done

Line 11 replace "have been" with "were" → done

Line 13 "temperature" appears twice → corrected

**Page 5** no comments

**Page 6**

Line 23 replace "the particle" with "particles" → done

Line 28 replace "defined" with "fixed / constant" → done

Line 29 replace "are" with "were" → done

Line 30 replace "need" with "needed" → done

**Page 7**

Line 1 remove "the"; → done

add comma after "that" → done

Line 3 replace "use" with "used"; → done

replace "the profiles" with "profiles of" → done

Line 6 replace "have been" with "were" → done

Line 11 replace "An other" with "Another" → done

Line 12 remove "have" → done

Line 25 duplication of "of" → corrected

Line 30 add comma after "critical" → done

**Page 8**

Line 1 replace "RSs" with "RS"; → done

replace "allows" with "allowed" → done

Line 10 add comma after "retrievals"; → done

replace "has been" with "was" → done

Line 12 replace "has been" with "was"; → done

replace "treated" with "discussed"; → done

change format of citation to "(Martucci et al., in preparation)" → done

Line 13 add comma after "UTC)" → done

Line 19 add comma after "UTC)" → done

**Page 9**

Line 1 remove "the" → done

Lines 7-10 I find this difficult to follow. Please rephrase. Commas are your (and your readers') friends!

For example: "In any case it is important to point out the good accomplishment of Ralmo in retrieving RH information. This can be concluded from this intercomparison, in which Ralmo showed very small biases and standard deviations (below 9%RH), which are indicative of accuracy and precision, respectively, of our measurements."

We agree with the comment of the referee and the proposed text.

Line 12 add comma after "studies" → done

Line 15 add comma after "2017"; → done

rephrase the rest of the sentence, for example: "…with smoke particles present during one, and mineral dust during the other." → done

Line 16 replace "in" with "of" → After the referee's suggestion we have decided to change the title of the subsection 5.1 to " **Case I: hygroscopic growth of smoke mixture**" and also subsection 5.2: **"Case II: hygroscopic growth of mineral dust particles"**

Line 18 replace "pannel" with "panel" → done

Line 19 replace "planetary" with "atmospheric" → done

Line 20 replace "PBL" with "ABL", and change throughout the manuscript → done

Line 21 replace "pattern" with "development"; → done

replace "along" with "during" → done

Line 24 replace "is" with "was"; → done

replace "pannel" with "panel" → done

Line 25 replace "for" with "during" or "in" → done (replace by "in")

Line 31 add "(blue colour map)" after "concentrations"; → done

replace "over" with "at" (you refer to surface concentration) → done

Line 32 replace "from ground to 3 km asl" with "from 1 to 3 km asl" → After removing the lowest back-trajectory of Fig 4b we have replaced that part of the sentence by "**… in the lower layers of the troposphere …**"

**Page 10**

Line 2 replace "for" with "in" → done

Line 10 remove "burning", this is already covered by "combustion" → done

Line 12 Please specify "that days" (or rather "those days"). Do you mean "in the period from 6th to 8th September"? → done

Line 14 replace "concentrations" with "concentration"; → done

replace "were" with "was"; → done

remove "than" → done

Line 15 replace "took place" with "occurred" → done

Line 16 add comma after (AAE) → done

Line 20 replace "of" with "over" → done

Line 24 replace "is" with "was" → done

Lines 24-25 remove either "Simultaneously to this increase," or "for the same layer"; → done

remove "that there is also"; → done

replace "can observe" with "observed"; → done

replace "moving" with "increasing" → done

Line 28 add comma after "respectively)"; → done

replace "is" with "was" → done

Line 30 replace "have been" with "were" → done

Line 31 add comma after "RH"; → done

remove "what is" → done

Line 34 replace "respect its" with "with respect to its" → done

**Page 11**

Line 1 replace "took a value of" with "was" → The sentence has been rephrased in the reviewed manuscript as: "**This parameter is proportional to the aerosol hygroscopicity and it had values of 0.48±0.08 at 355 nm and0.29±0.08 at 1064 nm.**"

Line 11 replace "have been" with "was" → done

Line 13 remove "have" → done

Line 21 replace "how" with "that"; → done

remove 2nd "the"; → done

replace 2nd "and" with comma → done

Line 22 add: "where D is particle diameter" after the equation → done

Line 29 remove "totally" → done

Line 32 add comma after "cases"; → done

remove "also means that we" → done

Line 33 replace "the theory of Mie" with "Mie theory"; → done

replace "," with "."; → done

replace "so" with "Hence," (start new sentence) → done

**Page 12**

Line 2 replace "For" with "In"; → done

replace "we can observe that r (Fig.9, top) is quite constant (…)" with "we can observe that r was quite constant (…, Fig. 9, top)" → done

Line 4 remove "The" → done

Line 6 remove "again" → done

Line 9 remove "very" → done

Line 10 replace "suffered by this aerosol" with "that occurred within this aerosol layer" → done

Line 12 remove "very"; → done

remove "either" → done

Lines 14-15 it sounds like you used a hygroscopic parameter of 0.4 for the later period (16:00 to 23:30); please rephrase; → Done. The sentence read now as (page 12, lines: 3-5): "**The humidogram obtained for this dehydration process (Fig.10a, blue open circles) also showed $f_\beta$ values very close to the ones calculated in the later period and a hygroscopic parameter from Hänel parametrization of 0.40±0.08**"

remove "also" in line 15 → done

Line 21 replace "pannel" with "panel" → done

Line 22 explain acronyms "NMMB" and "BSC" → done

Line 23 add comma after "Europe" → done

Line 24 replace "in" with "for" or "above" → done

Line 27 replace "in our station" with "at our station" → done

Line 29 replace "along" with "throughout" → done

Line 30 replace "along" with "during" or "throughout" → done

Line 31 remove parentheses and include as full sentence → done

Line 32 replace "is" with "was"; → done

add comma after "measurements"; → done

replace "analyze" with "analyzed" → done

Line 33 replace "observe" with "observed" → done

**Page 13**

Lines 5-6 put parenthesis and dot on same line as 0.24 (remove space?) → done

Lines 6-7 rephrase, for example:

As in case 1, we found the opposite spectral dependency compared to Lv et al. (2017).

However, we considered a wider spectral range. → done

Line 10 remove "also"; → done

remove "the" → done

Line 13 remove first "the" → done

Line 18 replace "16.6" with "15.6" (check if table 1 or text is correct); → done. It was a typo. The correct number is 15.6.

remove first "the" → done

Line 23 remove "the" → done

Line 24 remove first occurrence of "model" → done

Line 28 add "and relative" after "absolute" → done

Line 31 remove "the one of" → done

**Page 14**

Line 13 replace "monitor" with "observe" or "detect"; "monitor" sounds more like an automated process → done

Line 14 replace "in almost a continuous way" with "almost continuously" → done

Line 16 remove "the"; → done

replace "the particle" with "particles" → done

Line 17 add comma after "growth"; → done

replace "have been" with "were" → done

Line 21 replace "along" with "throughout" → done

Line 27 replace "in the full" with "throughout the" → done

Line 33 swap order of sentence: "…in the lower troposphere (...) were very similar to the ones obtained during nighttime." → done

**Page 15**

Line 14 add: "as well as the study of Haarig et al. (2017)" → done

Line 20 remove "very" → done

Line 21 add comma after "2017)"; → done

add comma after "particles" → done

Lines 26-27 remove first "the"; → done

remove "It"; → done

move "was also remarkable" to the end of the sentence → done

Line 28 replace "has been" with "was" → done

Line 30 remove first "the" → done

Line 31 add "and relative" after "absolute" → don

Line 33 replace "was not" with "not having been"; → done

add comma after "Therefore" → done

Line 34 replace "have" with "has" → done

**Page 16**

Line 6 replace "AA" with "AH"? → done

**Page 21**

Table 1 replace "layer width" with "layer depth" → done

**Page 22**

Figure 1 start caption with: "Mixing ratio (r), temperature (T) and relative humidity (RH) [...]" → done

Figure 2 add date after "UTC" → done

**Page 23**

Figure 3 replace "of" with "of vertical profiles of";

In the caption "r" is called "water vapour mixing ratio", and in the figure "WVPR".

Please amend figure and caption to "mixing ratio" and "r" to keep it consistent

throughout the manuscript and to avoid confusion. → done

**Page 24**

Figure 4 replace "Total Optical Depth" with "total optical depth from sulfate (orange/red scale),

dust (green/yellow scale), and smoke (blue scale)"; → done

possibly replace "1500, 2500, 3500 m asl" with "1000, 2000, 3000 m asl"; Payerne

being roughly at 500 m asl. 3500 m asl is not consistent with the text either (page 9,

line 32). → We have removed the lowest back-trajectory because it was less reliable being so close to ground.

**Page 26**

Figure 6 explain in caption what the grey shaded area means; → done

add "temperature" after "potential" → done

**Page 27**

Figure 8 increase font size in legends → done

**Page 28**

Figure 9 add "of" after "Evolution" → done

Figure 10 replace "Hannel" with "Hänel" in figure legends → done

**Page 29**

Figure 12 replace "pannel" with "panel" (3 times in this caption); → done

increase size of the three panels on the left; font sizes in left and right plot should be

the same -> → done

---

## Author Comment (AC2) · 23 Jul 2019

**Reviewer 2**

Review for manuscript "Towards continuous monitoring of aerosol hygroscopicity by Raman lidar measurements at the EARLINET station of Payerne"

Authors study important problem of the aerosol hygroscopic growth basing on long term multiwavelength lidar observations. The research is done on high scientific level. Authors well understand all the issues, when the information about humidification process is extracted from lidar measurements. Manuscript is well written and can be published after minor revisions. The Reviewer 1 provided very detailed review, so I can add just several technical comments.

Comment 1:

Title. I agree with Reviewer 1 that title can be shortened.

According to the suggestions of both referees we have modified the title of this paper. It reads now as:

**"Characterization of aerosol hygroscopicity using Raman lidar measurements at the EARLINET station of Payerne"**

Comment 2:

p.10, ln 24. "From this figure, a marked increase of with altitude is observed for the altitude range between 1.7 and 2.3 km (asl)." The same time for range 1.5-2.0 backscattering doesn't increase significantly, though RH rises. Any ideas why (potential temperature and mixing ratio are quite stable)?

The fact that the aerosol backscatter coefficients do not increase in the lower part of the layer (1.5-2.0), despite of having good mixing conditions (almost constant values of potential temperature and mixing ratio), is due to the lower RH at that range. For this example, the deliquesce relative humidity (DRH, RH at which solid particle spontaneously absorbs water) was around 80%RH which was reached at 2 km (asl).

Comment 3:

Fig.7. What will happen with these curves and Hanel parameters if starting height is 1.5 km? How sensitive are results to the choice of height interval?

We selected the range 1.7-2.3 km due to the higher stability of potential temperature and water vapour mixing ratio found in that range (mean($\theta$)=300.6±0.5 K; mean(r)= 5.3±0.1 g/kg), which indicated very good mixing conditions. However, following the referee's suggestion we have calculated the Hänel parameter for the layer 1.5-2.3 km, observing that its value ($\gamma$=0.47±0.07) is almost the same than the one obtained in the thinner layer (layer 1.7-2.3 km: $\gamma$=0.48±0.08). This result shows that the selection of our height interval was not very critical and give consistency to our results.

Comment 4:

Fig.11-13. Figures should be done in the same style: size, format, fonts, grids should be kept the same. Some fonts are very small, difficult to read. Probably Fig11b,c can be shown on the same plot.

The mentioned figures have been improved in the revised manuscript.

---

## Author Response (AR2)

**Authors' response**

We thank the anonymous reviewer for this last revision and propose the next technical corrections which have been implemented in the revised manuscript.

Page 9, Line 31: Replace "the days 7 and 8 of" with "7th and 8th".

Page 10, Line 23: Replace "humidity increases" with "humidity increased".

Page 13, Line 3: Replace "vary" with "increase".

Page 13, Line 4: Replace "lead" with "leads".

Page 13, Line 5: Replace "affect to the" with "affect the".

Figure 4b: Crop bottom part of this panel (starting with "Job ID"), it is not relevant and font is too small to read.

Figure 6 caption: Replace "occurs" with "occurred".